**Data Availability Statement:** https://github.com/p-shyam23/MOHO

**Funding:** The author(s) received no specific funding for this work.

# Optimal truss design with MOHO: A multi-objective optimization perspective

**Nikunj Mashru**[1], **Ghanshyam G. Tejani**[2], **Pinank Patel**[1], **Mohammad Khishe**[3,4,5]*

**1** Department of Mechanical Engineering, Faculty of Engineering and Technology, Marwadi University, Rajkot, Gujarat, India, **2** Ethics Infotech, Vadodara, Gujarat, India, **3** Department of Electrical Engineering, Imam Khomeini Naval Science University of Nowshahr, Nowshahr, Iran, **4** Innovation Center for Artificial Intelligence Applications, Yuan Ze University, Taoyuan City, Taiwan, **5** Applied Science Research Center, Applied Science Private University, Amman, Jordan

\* m_khishe@alumni.iust.ac.ir

## Abstract

This research article presents the Multi-Objective Hippopotamus Optimizer (MOHO), a unique approach that excels in tackling complex structural optimization problems. The Hippopotamus Optimizer (HO) is a novel approach in meta-heuristic methodology that draws inspiration from the natural behaviour of hippos. The HO is built upon a trinary-phase model that incorporates mathematical representations of crucial aspects of Hippo's behaviour, including their movements in aquatic environments, defense mechanisms against predators, and avoidance strategies. This conceptual framework forms the basis for developing the multi-objective (MO) variant MOHO, which was applied to optimize five well-known truss structures. Balancing safety precautions and size constraints concerning stresses on individual sections and constituent parts, these problems also involved competing objectives, such as reducing the weight of the structure and the maximum nodal displacement. The findings of six popular optimization methods were used to compare the results. Four industry-standard performance measures were used for this comparison and qualitative examination of the finest Pareto-front plots generated by each algorithm. The average values obtained by the Friedman rank test and comparison analysis unequivocally showed that MOHO outperformed other methods in resolving significant structure optimization problems quickly. In addition to finding and preserving more Pareto-optimal sets, the recommended algorithm produced excellent convergence and variance in the objective and decision fields. MOHO demonstrated its potential for navigating competing objectives through diversity analysis. Additionally, the swarm plots effectively visualize MOHO's solution distribution of MOHO across iterations, highlighting its superior convergence behaviour. Consequently, MOHO exhibits promise as a valuable method for tackling complex multi-objective structure optimization issues.

**Competing interests:** The authors have declared that no competing interests exist.

## 1. Introduction

Engineers dealing with structural design often have competing objectives that can adversely affect each other. For instance, they might attempt to reduce the structure's weight for economic purposes, even if it would increase safety by strengthening the structure. MO optimization problems are characterized by multiple objectives, requiring mathematical optimization strategies to deal with them effectively. Multi-objective optimization problems typically need more than simple solutions. Instead, their responses usually manifest as a series of optimal solutions embodying a balance between the conflicting objectives being optimized. This trade-off guarantees that achieving a particular purpose will inevitably result in the degradation of other objectives [1, 2]. The Pareto front provides the designer with various feasible alternatives by meeting all the constraints; among these, the final design can be selected. Applying meta-heuristic optimization methods has become an increasingly popular research direction for multi-objective structural design. When dealing with difficult circumstances, metaheuristic algorithms have become highly sophisticated techniques concentrating on minimizing or maximizing an objective function to arrive at an optimal solution. These algorithms efficiently traverse the solution space by utilizing upper-level searching techniques.

Recently developed new metaheuristics such as the Synergistic Swarm Optimization algorithm. [3], Geyser Inspired Algorithm [4], Zebra Optimization Algorithm [5], Quadratic Interpolation Optimization [6], Serval Algorithm [7], Egret Swarm Optimization Algorithms [8], Waterwheel Plant Algorithm [9], Propagation Search Algorithm [10], Mantis Search Algorithm [11], Komodo Mlipir Algorithm [12], Eik herd optimizer [13], Crayfish optimization algorithm [14], Kepler optimization algorithm [15], Light Spectrum Optimization [16], Circle Search Algorithm [17], Electric eel foraging optimization [18], Puma optimizer [19], Partial reinforcement optimizer [20], The coronavirus search optimizer [21], geometric mean optimizer [22], Fick's law algorithm [23], Prairie dog optimization algorithm [24]. The Arithmetic optimization algorithm [25], Grasshopper optimization algorithm [26], inspired by natural mechanisms, including evolutionary processes, physical lows, Mathematical theories, and behaviour observed in animals.

Numerous researchers have introduced innovative optimization algorithms across various applications, prioritizing statistical validation and rigorous experimentation to enhance the convergence time and solution quality compared to existing methods. These algorithms address complex optimization problems in domains such as engineering design. [27], meta-heuristic approaches, and medical diagnostics [28], demonstrating encouraging developments in effectively resolving practical issues [29] and different strategies [30, 31], such as boundary update techniques, multi-hybrid approaches [32], multi-strategy infused metaheuristics [33–35]. Each algorithm incorporates new metaheuristic methodologies to handle specific optimization goals and enhance overall performance.

These approximate optimization methods provide sufficiently good solutions for complex situations within a reasonable time frame. Population-based metaheuristics are particularly noteworthy since they are effective tools that may be used to solve MO optimization problems [36]. Some notable multi-objective algorithms include the MO version of Atomic Orbital Search. [37], MO version of the Material Generation Algorithm [38], MO Crystal Structure Algorithm [39], MO Chaos Game Optimization [40], NSGA-2 [41], MO Arithmetic Optimization Algorithm [42], MO Thermal Exchange Optimization [36], MO Passing Vehicle Search Algorithm [43], MO symbiotic organism search [44], and MO heat transfer search [45]. Many researchers have Improved MO optimization algorithms with unique approaches, such as decomposition-based MO heat transfer search. [46], improved MO particle swarm optimization [47], an indicator-based multi-SSM algorithm [48], MO improved marine predators

algorithm [49], MO structural optimization using improved heat transfer search [50] Enhanced MO GWO with levy flight and mutation operators for feature selection [51], and a two-archive MO multi-verse optimizer for truss design [52].

The primary approach to estimating a Pareto front combines a non-dominated (ND) sorting technique with the population-based concept of meta-heuristics in MO optimization. The Pareto archive is continually refined by updating it with data from the current population and the archive of the previous iteration through the iterative reproduction of design solutions. This process was repeated until the Pareto front of the design problem was reached. Critical performance factors are optimizing the parameter settings using a self-adaptive method and preserving the search diversity through efficient clustering approaches. The main intention is to significantly increase the search intensity and variety because MO metaheuristics require extensive design space exploration while maintaining a high convergence rate, which is a more difficult challenge than single-objective metaheuristics.

The "No Free Lunch" (NFL) [53] theorem serves as a reminder that no individual metaheuristic is universally capable of solving all real-world problems. This understanding has catalyzed the advancement and enhancement of diverse metaheuristic methods. This research introduces an MO version of the Hippopotamus Optimization algorithm (HO) [54], drawing inspiration from the observed behaviors in hippos. Emphasizing traits and actions, such as their semi-aquatic lifestyle, herbivorous diet, and defensive strategies against predators, the algorithm incorporates their formidable jaws and warning vocalizations. Furthermore, it explores the adaptability of protective behaviors and social dynamics, aiming to understand their applicability in various contexts through insights from the hippopotamus behavior.

It is interesting to evaluate the performance of a newly created multi-objective metaheuristic in various engineering design challenges. This study applies MOHO to different five-truss structures: 10-bar truss, 25-bar truss, 60-bar ring truss, 72-bar truss, and 942-bar tower truss. The design challenge is decreasing the structure's mass while decreasing the maximum nodal displacement, a structural stiffness indicator. The above objectives were pursued within the limitations of area and stress. HO's mathematical model for organizing hippopotamuses within herds considers factors such as dominance and proximity to optimize their positioning in aquatic environments. defensive strategies for hippos against predators, including vocalizations and tactical movements, to ensure herd safety. Additionally, it outlines a behavior in which hippos escape predators by seeking refuge in water bodies, enhancing their survival prospects. Overall, this study explored methods to improve the safety and survival of hippopotamus herds through mathematical modeling and behavioral analysis. The primary progress of this investigation and its evolution, which surpasses the present contemporary, are outlined as follows.

1. A multi-objective version of the unique hippopotamus optimization algorithm (MOHO) was applied to five planar and spatial truss structures to minimize the maximum nodal displacement and structural mass subjected to area and stress constraints.

2. The performance of MOHO for different truss structures was compared with six well-known and efficient optimization algorithms, viz., MO Ant System (MOAS) [55], MO Ant colony system (MOACS) [56], MO differential evolution (DEMO) [57], NSGA-2 [41], MO ant lion optimizer (MOALO) [58, 59], and MO moth flame optimizer (MOMFO) [60].

3. Four commonly used performance metrics were used to assess the algorithms' efficacy statistically: the Hypervolume Index (HV), Generational Distance (GD), Inverted Generational Difference (IGD), and spacing-to-extent (STE). In addition, each algorithm's best Pareto-front plots were closely examined to evaluate the qualitative behavior. In addition,

the algorithms were ranked for a thorough study using Friedman's test at the 95% significance level.

4. The results presented a fresh outlook on the benefits and drawbacks of MO optimization techniques in addressing diverse contradicting objectives in structural optimization.

The outline of the paper is as follows:

▪ The 2nd section presents a description and mathematical modeling of the elementary Hippopotamus Optimization Algorithm (HO).

▪ 3rd section elaborates on the proposed MOHO and outlines the formulations of MO structure optimization problems.

▪ In the 4th section, an experimental assessment of the MOHO optimizer and a comparison with other prominent algorithms for addressing truss bar problems are presented.

▪ Section 5 discusses the performance metrics and enhanced Pareto fronts using diversity curves and swarm plots.

▪ Finally, Section 6 concludes the study and discusses future work to explore the capability of MO algorithms.

## 2. Hippopotamus optimization algorithm

The hippopotamus, a captivating African vertebrate mammal, thrives in semi-aquatic habitats, such as rivers and ponds, displaying social behaviors within pods. Despite the challenges in gender determination, their herbivorous diet and exploratory nature drive them to investigate alternative food sources [61]. They are deemed one of the most dangerous mammals with immense strength and territorial behaviour, yet predators avoid confronting adult hippos because of their size. Defensive strategies include aggressive postures, loud vocalizations, and the rapid retreatment of water bodies. Inspired by the observed behavioral patterns, hippopotamus groups consist of various members, with calves prone to wandering and becoming prey [62]. Defensive behaviors include rotation toward predators, employing powerful jaws, and fleeing toward water sources for safety [63]. This algorithm uses two exploration stages and one exploitation phase after initial random solutions are produced. These phases are perceived to be superior to other algorithms for truss design problems.

### 2.1 Exploration phase-1 (position update in river or pond)

A herd of hippos consists of adult females, calves, adult males, and a dominant male leader. A value iteration process establishes dominance. While females encircle the dominant male, they defend the herd and its area. When a male reaches adulthood, the dominant male drives him out, forcing him to fight for supremacy elsewhere. Table 1 shows the mathematical positions of the hippos and their updates in the herd's habitat.

Stage 1 in Table 1 represents the population of hippos after initialization within the upper and lower bounds of the lake or pond. Stage 2 shows five different scenarios based on random numbers $h_1$ and $h_2$. Stages 3 and 4 demonstrate the position of immature hippos from their mothers within the herd, otherwise separated based on the T value. Finally, in Stage 5, the position is updated based on its objective value.

**Table 1. Stages of exploration phase-1.**

| Stage | Mathematical Representations | Description |
|---|---|---|
| 1 | $P_i^{male-hippo} : p_{i,j}^{male-hippo} = P_{i,j} + y_1 * (D_{dominant-hippo} - I_1 p_{i,j})$ <br> $for, i = 1, 2, \ldots, \left[\frac{N}{2}\right] and, j = 1, 2, \ldots, m$ | $P_i^{male-hippo}$ is the position of a male hippo. <br> $D_{do\ min\ ant-hippo}$ is the position of the dominant hippo. <br> i = candidate solutions. <br> j = decision variables. <br> $y_1 = [0,1]$. <br> $I_1 = [1,2]$. <br> N = population size |
| 2 | $h(senario) = \begin{cases} I_2 * \vec{r}_1 + (\sim Q_1) \\ 2 * \vec{r}_2 - 1 \\ \vec{r}_3 \\ I_1 * \vec{r}_4 + (\sim Q_2) \\ \vec{r}_5 \end{cases}$ | $\vec{r}_1, \vec{r}_2, \vec{r}_3, \vec{r}_4, \vec{r}_5 = [o,1]$. <br> $I_1, I_2 = [1,2]$. <br> $Q_1$ and $Q_2$ = 0 or 1. |
| 3 | $T = \exp(-\frac{t}{T})$ <br> $P_i^{Female-hipppo} : p_{i,j}^{Female-hippo} = \begin{cases} p_{i,j} + h_1 * (D_{dominant-hippo} - I_2 Mgi)\ if, T > 0.6 \\ E \qquad\qquad\qquad\qquad\qquad else \end{cases}$ | $Mgi$ = mean of some randomly selected hippos. <br> $T$ represents the distance of an immature hippo from its mother. |
| 4 | $E = \begin{cases} p_{i,j} + h_2 * (Mgi - D_{dominant-hippo})\ \ if, r_6 = 0.5 \\ Lb_j + r_7 * (Ub_j - Lb_j) \qquad\qquad else \end{cases}$ <br> $for, i = 1, 2, \ldots, \left[\frac{N}{2}\right] and\ j = 1, 2, \ldots, m.$ | $P_i^{Female-hipppo}$ = female or immature hippos. $h_1$ & $h_2$ are selected from the equation of h. $r_7 = [0,1]$ |
| 5 | $P_i = \begin{cases} P_i^{male-hippo} \quad if, F_i^{male-hippo} < F_i \\ P_i \qquad\qquad else \end{cases}$ | Male and female hippos' positions update within the herd. $F_i$ is an objective function value. |

## 2.2 Exploration phase-2 (defense against predators)

In this phase, hippopotamuses find safety in their herds, deterring predators due to their large size and collective presence. However, young and sick individuals are more susceptible to predator attacks. When threatened, hippopotamuses emit loud vocalizations and may approach predators to avoid potential threats. Table 2 demonstrates a mathematical representation of half of the population of hippos with exploration phase 2.

In the second phase of exploration, which spans stages 6 to 11, the algorithm simulates the defense mechanisms of hippos against predators.

In Stage 6, the positions of the predators were determined within the search space. At stage 7, the algorithm evaluates the position of each predator relative to each hippo, triggering the corresponding reactions from the hippos. Stage 9 involves random movements of hippos, influenced by a Levy distribution, allowing search space exploration. Stages 10 and 11 determine whether predators hunt the hippos or if they successfully escape from predators. Overall, this phase simulates a dynamic interaction between predators and hippos, guiding the exploration process and helping to prevent the algorithm from becoming trapped in local minima.

## 2.3 Exploitation phase (escaping from the predator)

In this exploitation phase, as shown in Table 3, hippos typically run to the closest body of water for safety when separated from the herd and attacked by lions or spotted hyenas. This strategy improves local search capabilities by simulating the habit of seeking refuge nearby. This behavior involves creating a random site close to the hippopotamus's location to increase

**Table 2. Stages of exploration phase-2.**

| Stage | Mathematical Representations | Description |
|---|---|---|
| 6 | $Predator : Predator_j = Lb_j + \vec{r}_b*(Ub_j - Lb_j), \; j = 1, 2, \ldots, m.$ <br><br> $\vec{D} = \lvert Predator_j - P_{i,j} \rvert$ | Predator's position in the search space. <br> $\vec{D}$ is a distance of i$^{th}$ hippo to the predator. <br> $\vec{r}_b$ = random [0,1] |
| 7 | $P_i^{R-hippo} : P_{i,j}^{R-hippo} = \begin{cases} \overrightarrow{RL} \oplus Predator_j, +(\frac{\hbar}{(c - d*\cos(2\pi g))})*(\frac{1}{\vec{D}}) & F_{predator_j} < F_i \\ \overrightarrow{RL} \oplus Predator_j, +(\frac{\hbar}{(c - d*\cos(2\pi g))})*(\frac{1}{2*\overrightarrow{D + r_9}}) & F_{predator_j} \geq F_i \end{cases}$ | Defensive behavior of hippos based on the factor $F_{predator_j}$ for protection against predators. <br> $\overrightarrow{RL}$ is a random vector based on levy distribution. |
| 8 | $for, \; i = [\frac{N}{2}] + 1, [\frac{N}{2}] + 2, \ldots, N \; and \; j = 1, 2, \ldots, m$ | |
| 9 | $Levy(v) = 0.05*\frac{w*\sigma_w}{\lvert v \rvert^{\frac{1}{v_1}}}$ | $w$ and $v$ are random numbers in [0,1]. <br> $v_1 = 1.5$ |
| 10 | $\sigma_w = [\frac{\Gamma(1+v_1)\sin(\frac{\pi v_1}{2})}{\Gamma(\frac{(1+v_1)}{2})v_1*2^{\frac{(v_1-1)}{2}}}]^{\frac{1}{v_1}}$ | $\Gamma$ = gamma function and $\hbar$ is [2,4], c = [1,1.5], d = [2,3], g = [−1,1]. <br> $r_9$ = is a random vector with 1*m |
| 11 | $P_i = \begin{cases} P_i^{R-hippo} & if, \; F_i^{R-hippo} < F_i \\ P_i & F_i^{R-hippo} \geq F_i \end{cases}$ | If hippos have been hunted, other hippos will replace them in the herd; otherwise, the hunter will escape, and the hippos will return to the herd. |

the cost function value. Iteratively, the hippopotamus adjusts its position to guarantee its proximity to safety.

According to Table 3, during the exploitation phase, hippos can use the value of S from Stage 14 to determine a safe location near where they are right now to protect themselves from predators. Finally, a position update for the hippos was made using their objective values.

# 3. Multi-objective (MO) problem formulation

## 3.1 MO optimization definitions

In optimizing truss structures with multiple objectives, the focus is on simultaneously achieving optimization across various goals. While the term "multi-objective" typically addresses problems involving up to three objectives, the emergence of "many-objective optimization" many-objective optimization tackles the challenges posed by numerous objectives. Resolving conflicts between objectives is a significant challenge in multi-objective optimization, requiring specific approaches. The traditional relational comparisons between solutions are inadequate when there are several criteria. Therefore, alternative operators, such as the Pareto

**Table 3. Stages of the exploitation phase.**

| Stage | Mathematical Representations | Description |
|---|---|---|
| 12 | $Lb_j^{local} = \frac{Lb_j}{t}$ <br><br> $Ub_j^{local} = \frac{Ub_j}{t}, \; t = 1, 2, \ldots, T_{max}$ | t is the current iteration, and T is the maximum iteration. |
| 13 | $P_i^{E-hippo} : P_{i,j}^{E-hippo} = p_{i,j} + r_{10}*(Lb_j^{local} + S(Ub_j^{local} - Lb_j^{local}))$ <br><br> $i = 1, 2, \ldots, N, j = 1, 2, \ldots, m.$ | $P_i^{E-hippo}$ is the position of the hippo to find a closet safe place. <br> $r_{10} = [0,1]$. |
| 14 | $S = \begin{cases} 2*\vec{r}_{11} - 1 \\ r_{12} \\ r_{13} \end{cases}$ | S leads to more suitable local search (exploitation). <br> $\vec{r}_{11} = [0,1]$, $r_{13} = [0,1]$. <br> $r_{12}$ is normally distributed random number. |
| 15 | $P_i = \begin{cases} P_i^{E-hippo} & if, \; F_i^{E-hippo} < F_i \\ P_i & F_i^{E-hippo} \geq F_i \end{cases}$ | $P_i$ = Position of hippos. |

dominance operator, are essential to assess the relative superiority. From a mathematical point of view, Pareto optimality defines a set of solutions deemed nondominated and optimal for a given problem, particularly for MO optimization. These solutions constitute the MO optimization solution set, which thoroughly depicts realistic trade-offs between several objectives. This set, which illustrates the optimal solutions possible within the objective domain, is frequently represented and visualized as a Pareto optimal front (PF). Further clarification regarding the principles of domination and associated ideas can be obtained from Fig 1, which graphically presents these concepts.

## 3.2 Multi-Objective Hippopotamus Optimization algorithm (MOHO)

The MOHO algorithm starts with a random population (hippos) size $N$. It generates solutions $X_1, X_2, \ldots, X_N$ within the specified search space defined by the lower and upper bounds of the problem for each dimension ($D$). The fitness function evaluates and assigns a fitness score to each candidate solution, guiding the selection process toward the optimal solutions. The fitness of each candidate solution was assessed using the fitness function provided for all initial solutions. The main loop of the HO algorithm starts with the first iteration and continues up to a specified number of generations. The best candidate solution (Xbest) and its corresponding fitness (fbest) were updated for each iteration. The loop was divided into three stages (two exploration phases and one exploitation phase).

Half of the total population explores the search space in exploration phase 1; each individual calculates two potential positions ($X\_P_1$ and $X\_P_2$) based on its current position, the best solution found so far ($X_{best}$), and a randomly selected mean group ($MeanGroup$). The selection of potential positions involves randomness, which is affected by various parameters. Individuals

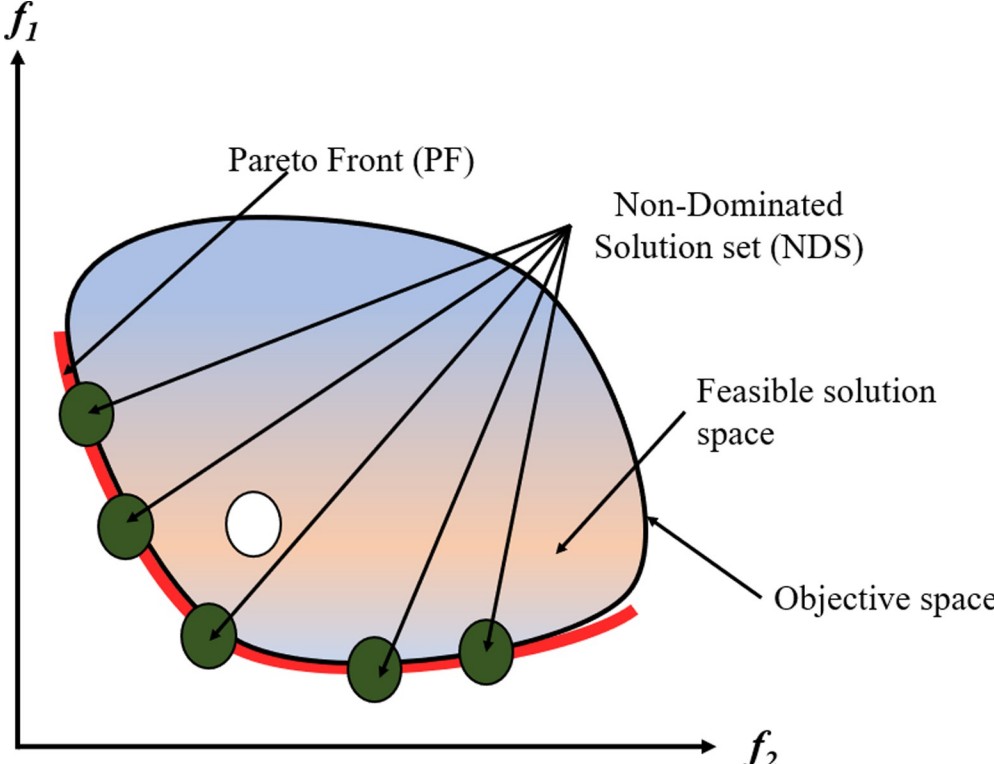

**Fig 1. MO problem definition.**

update their positions based on their potential positions if their fitness is improved. The remaining individuals in the population defend themselves against potential threats from predators. Each individual calculates a potential position ($X\_P_3$) based on its current position, randomly generated predator position, and Levy flight. The decision to update positions is based on whether the new position improves fitness compared to the current position. All individuals perform a localized search to escape potential predators and exploit promising regions. Each individual calculates a potential position ($X\_P_4$) based on its current position and randomly selected direction (*D*). Positions are updated if the new position improves fitness. The best solution found thus far, along with its corresponding fitness value, was stored for *M* objective functions as Pareto fronts. For every iteration, all fronts' non-domination (ND) sorting process helps achieve superior ND fronts.

Fig 2 shows a flow chart of MOHO with two exploration phases, dividing the total population by half size for phase 1 (position update in rivers or ponds) and half size for phase 2 (defense against predators). After passing through any of the exploration phases, phase 3 (escaping from predators), the exploitation phase, will be activated, and the positions of hippos will be updated accordingly. The evaluated solution at the end of this phase is the non-dominated Pareto optimal front.

## 4. Formulation of the truss design problem

MO truss design problems aim to determine the best design variables to minimize the structure's weight and optimize the nodal displacement while adhering to the tensile and compressive stress constraints. The computational problem is defined as follows.

$$\text{Find}, X = \{X_1, X_2, \ldots, X_m\} \tag{1}$$

*to minimize the structure mass*

$$f_1(X) = \sum_{i=1}^{m} X_i \rho_i L_i$$

& *to minimize maximum nodal displacement*

$$f_2(X) = max\left(|\delta_j|\right)$$

Subject to:

$$g_1(X) : \text{tensile \& compressive stress constraints}, |\sigma_i| - \sigma_i^{max} \leq 0$$

$$g_2(X) : \text{cross} - \text{sectional areas}, X_i^{min} \leq X_i \leq X_i^{max}$$

where, $i = 1,2, \ldots, m; j = 1,2, \ldots, n$

Here, *Ax* is a vector for the cross-sectional area; $\rho_i$ and $Ln_i$ are the mass density and the length of an element, respectively; *E* and σ are the Modulus of elasticity and stress, respectively.

The first objective is the mass of the structures and the movement of the nodes, or connection points, throughout the truss construction, and is called nodal displacement, which is the secondary objective. We can ensure that the structure will deflect sufficiently under loads without exceeding the critical limits by optimizing the nodal displacement.

We used dynamic penalty functions to manage the constraints associated with compressive and tensile stresses [64]. The penalty function penalizes the objective functions based on the

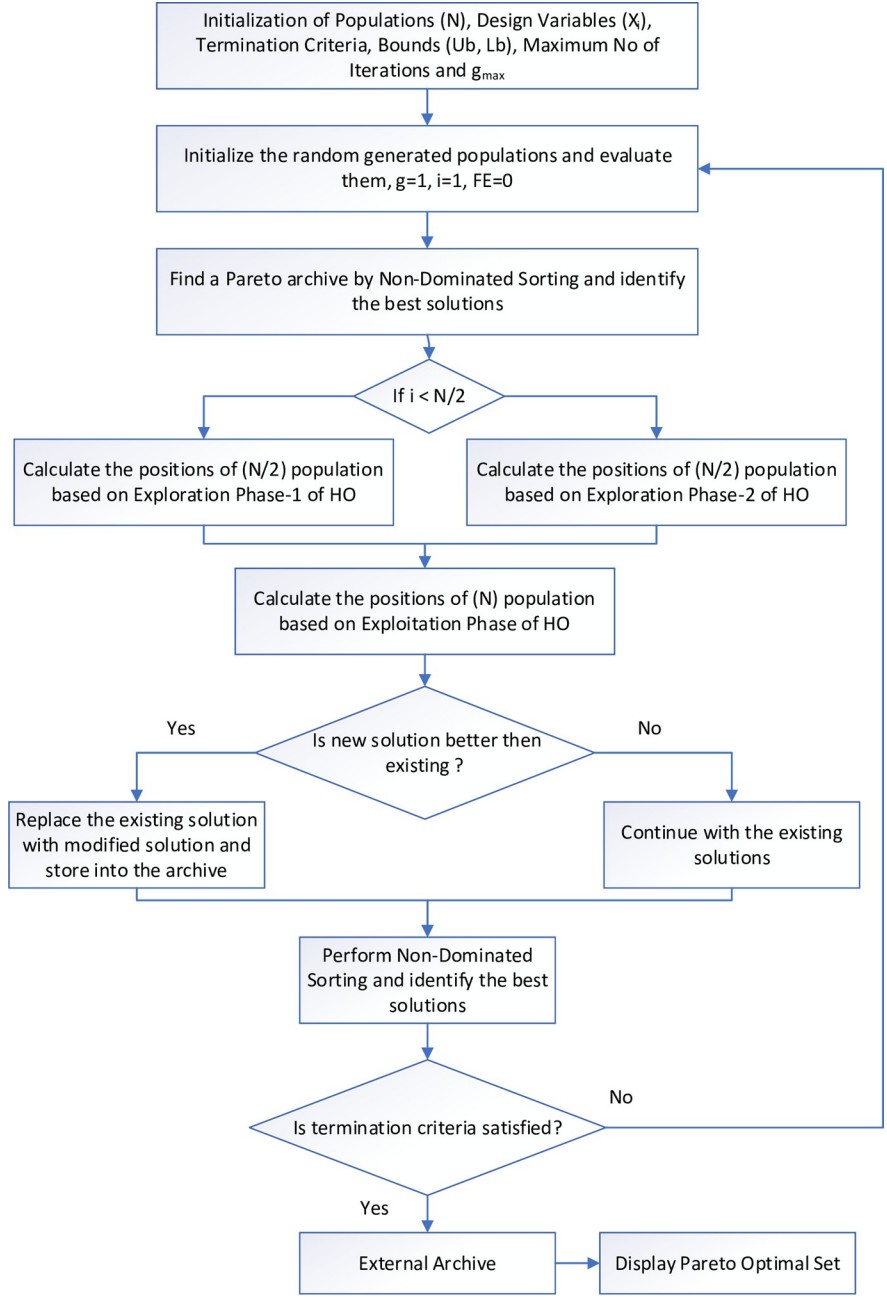

**Fig 2. Flow chart of the MOHO.**

extent of the constraint violation. The penalty function is formulated as follows:

$$f_{penalty}(X) = \begin{cases} f(X), & no\ constraint\ violation \\ f(X)*(1 + \varepsilon_1 * C)^{\varepsilon_2}, C = \sum_{i=1}^{q} C_i, C_i = |1 - \dfrac{p_p}{p_i^*}|, & otherwise \end{cases} \quad (2)$$

where, $p_i$ is the value of constraint violation concerning the bound $p_i^*$. C is a penalty function that calculates the total constraint violation, where Ci represents the extent of the breach for

each constraint. The penalty function adjusts the objective functions based on the total constraint violation, with a penalty factor determined by constants $\varepsilon_1$ and $\varepsilon_2$. The constants values, $\varepsilon_1$ and $\varepsilon_2$ are taken three based on comparable research findings from the literature [65]. The primary objective is to minimize the maximum nodal displacement and total mass by optimizing design variables using Finite Element Analysis (FEA). Stress and cross-sectional area constraints must be followed during this optimization process to assure structural integrity and safety.

### 4.1 Truss structural problems

Five standard benchmark truss problems—the 10-bar, 25-bar, 60-bar, 72-bar, and 942-bar trusses—are used to evaluate the performance of the studied algorithms. Furthermore, the published results and those from previous studies are compared [56]. The following sections provide details of these five common benchmark truss problems. Figs 3–7 show the investigated truss constructions and their loading circumstances, with geometrical dimensions used for computational evaluations. Tables 4–8 summarize the truss design considerations.

The first structural problem is a 10-bar truss, depicted in Fig 3. Angelo et al. have extensively studied this 2-D truss [56, 65–68]. Ten design variables were used to solve MO problems. As depicted in Fig 3, a vertical downside load of 100 kips is applied at nodes 2 and 4.

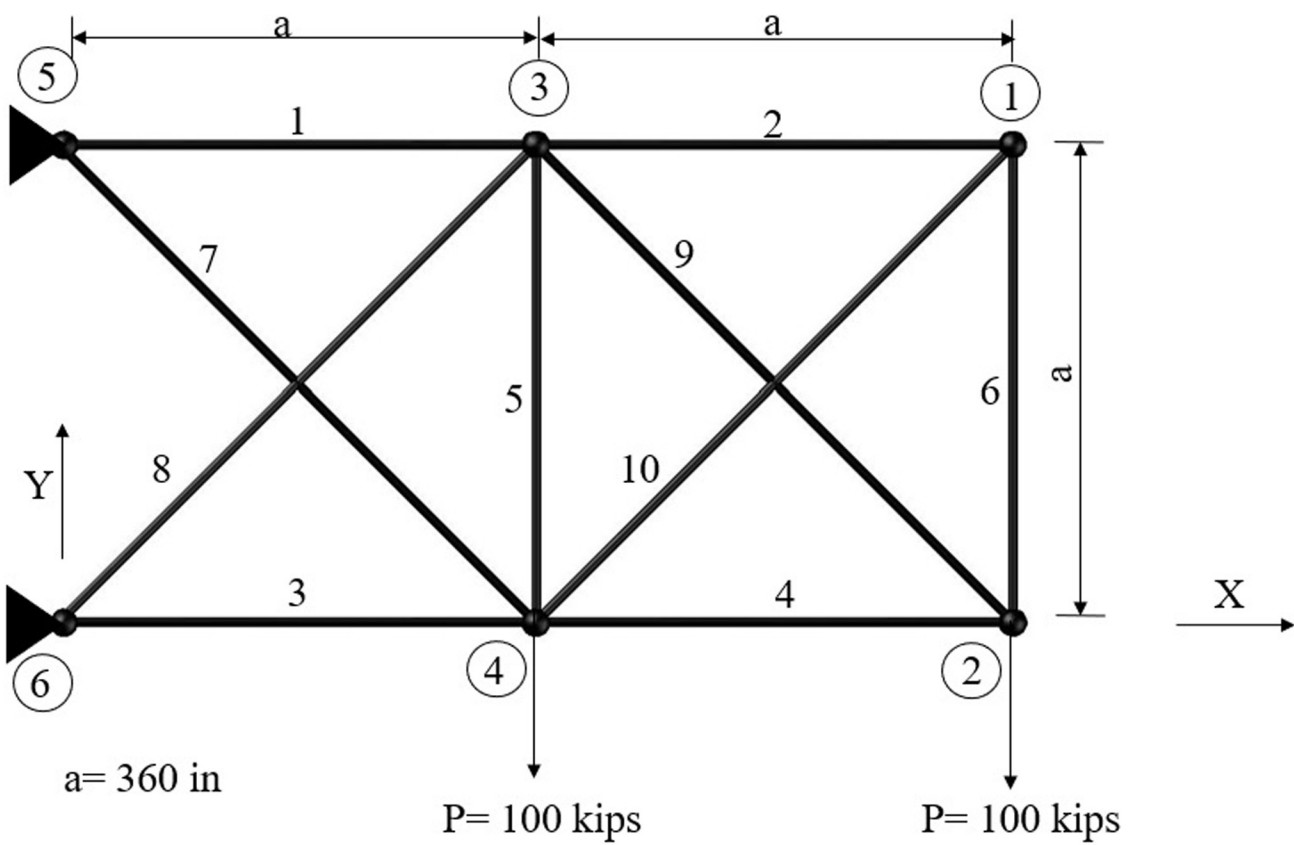

**Fig 3. The 10-bar truss structure.**

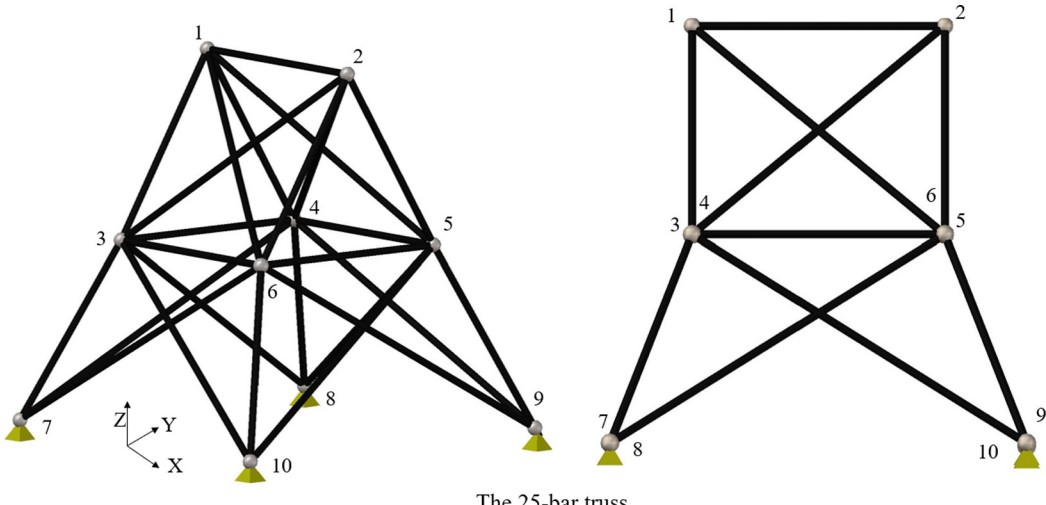

**Fig 4. The 25-bar truss structure.**

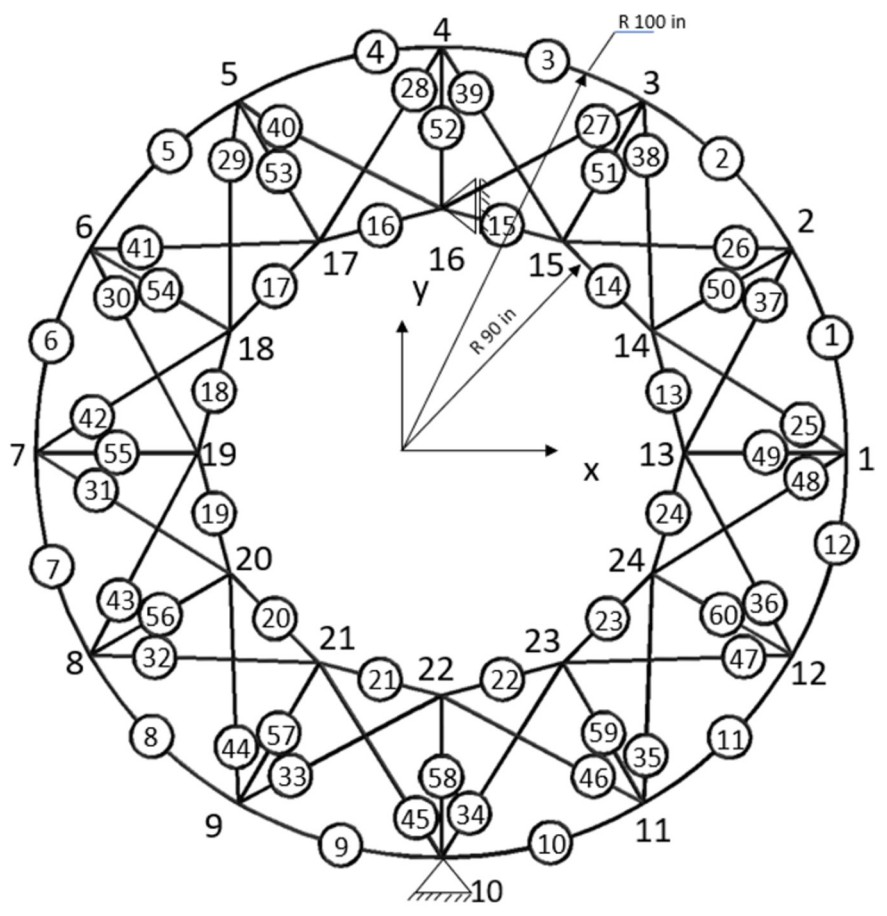

The 60-bar truss

**Fig 5. The 60-bar truss structure.**

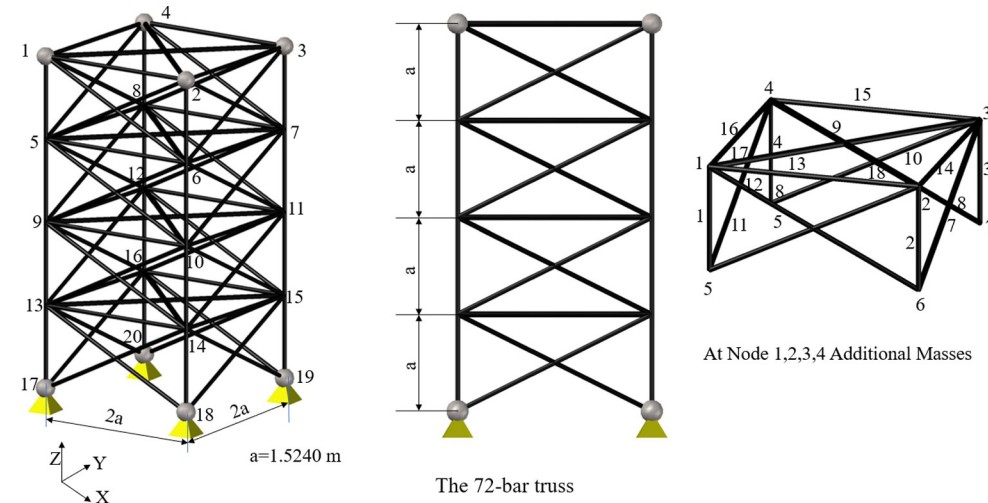

**Fig 6. The 72-bar truss structure.**

Fig 4 shows the 25-bar truss used in the second truss problem. Academic research regularly utilizes this 3-D truss [56, 66–68]. It is symmetric about the x-z and y-z planes, and its twenty-five elements are arranged into eight groups. Table 5 shows the density and Yong's modulus of the truss material with loading conditions at different nodes.

The third benchmark truss is a Ring truss with 60 bars. [56, 66–68], as seen in Fig 5. Table 6 illustrates how the truss is divided into 25 parts, each representing a symmetry. This truss is subjected to three load cases with size variable (cross-sections) values ranging from 0.5 in2 to 4.9 in2. Material properties are displayed in Table 6.

The 4th benchmark, which is a 72-bar 3-D truss [56, 66–68], is presented in Fig 6. As shown in Table 7, the truss is made up of 72 elemental cross-sections that are arranged into 16 segments. Two different load cases are considered as per Table 7. The maximum allowable stress is 25 ksi as a constraint to prevent structure failure. Material properties are as per Table 7 for the 3-D 72-bar truss structures. Here, 72-bar elements are grouped into 16 members as design

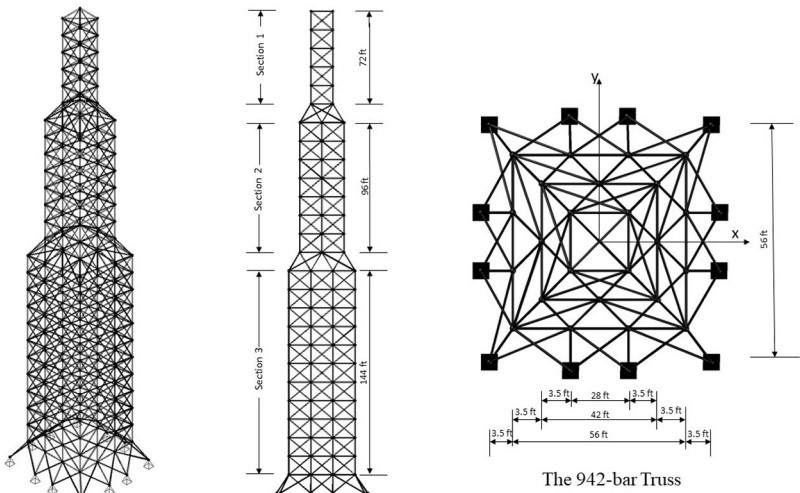

**Fig 7. The 942-bar truss structure.**

**Table 4. Design configuration of 10-bar problem.**

| The 10-bar Truss Structures | |
|---|---|
| Design variables | $X_i, i = 1,2,\ldots 0$ |
| Constraints ($\sigma^{max}$ in ksi) | 25 |
| Density ($\rho$ in lb/in$^3$) | 0.1 |
| Young modules (E in ksi) | 10000 |
| loading conditions (kips) | $P_{y2} = P_{y4} = -100$ |
| Grouping of Members | ———- |
| Size variables (in$^2$) | [1.62,1.8,1.99,2.13,2.38,2.62, 2.63,2.88,2.93,3.09,3.13,3.38, 3.47,3.55,3.63,3.84,3.87,3.88, 4.18,4.22,4.49,4.59,4.8,4.97, 5.12, 5.74,7.22,7.97,11.5,13.5, 13.9,14.2,15.5,16,16.9,18.8, 19.9,22,22.9,26.5,30, 33.5] |

variables. Size variables represent possible cross-sectional areas that can be assigned to truss structures ranging from 0.1 in$^2$ to 2.5 in$^2$.

As seen in Fig 7, the fifth and last truss is a large tower truss with 942 bars [44, 56, 67] To preserve symmetry, the 942 members of the truss geometry are divided into 52 members, representing design variables as per Table 8. Two loading conditions are given: nodal conditions and lateral loading, which are applied at different truss portions. Size variables for this huge tower range from 1 in2 to 200 in2. Material properties are density and Young's modulus, as per Table 8.

This study rigorously tested six different algorithms, with each approach undergoing one hundred runs for each studied truss problem. These evaluations were conducted meticulously, with each test encompassing 50,000 functional evaluations, ensuring a comprehensive analysis of algorithmic performance across various scenarios and challenges inherent in truss design problems.

## 4.2 Empirical assessment

MOHO was applied to evaluate all considered truss problems and ascertain the effectiveness of the approximate Pareto-optimal solutions generated by the MO optimization algorithm. The

**Table 5. Design configuration of 25-bar problem.**

| The 25-bar Truss Structures | |
|---|---|
| Design variables | $X_i, i = 1,2,\ldots,8$ |
| Constraints ($\sigma^{max}$ in ksi) | 40 |
| Density ($\rho$ in lb/in$^3$) | 0.1 |
| Young modules (E in ksi) | 10000 |
| loading conditions (kips) | $P_{x1} = 1,$ $P_{y1} = P_{z1} = P_{y2} = P_{z2} = -10,$ $P_{x3} = 0.5, P_{x6} = 0.6$ |
| Member Grouping | $X_1(1,2)$; $X_2(1–4, 2–3, 1–5, 2–6)$; $X_3(2–5, 2–4, 1–3, 1–6)$; $X_4(3–6, 4–5)$; $X_5(3–4, 5–6)$; $X_6(3–10, 6–7, 4–9, 5–8)$; $X_7(3–8, 4–7, 6–9, 5–10)$; $X_8(3–7, 4–8, 5–9, 6–10)$ |
| Size variables (in$^2$) | [1,.2,.3,.4,.5,.6,.7,.8,.9,1,1.1,1.2,1.3,1.4,1.5,1.6,1.7,1.8,1.9,2,2.1,2.2,2.3,2.4,2.5,2.6,2.8,3,3.2, 3.4] |

**Table 6. Design configuration of 60-bar problem.**

| The 60-bar Ring Truss Structures | |
|---|---|
| Design variables | $X_i, i = 1,2,\ldots 5$ |
| Constraints ($\sigma^{max}$ in ksi) | 40 |
| Density ($\rho$ in lb/in$^3$) | 0.1 |
| Young modules (E in ksi) | 10000 |
| loading conditions (kips) | Case 1: $P_{x1} = -10, P_{x7} = 9$ |
| | Case 2: $P_{x15} = P_{x18} = -8, P_{y15} = P_{y18} = 3$ |
| | Case 3:: $P_{x22} = -20$ *and* $P_{y22} = 10$ |
| Grouping of Members | $X_1(49–60); X_2(1,13); X_3(2,14); X_4(3,15); X_5(4,16); X_6(5,17); X_7(6,18); X_8(7,19); X_9(8,20);$ $X_{10}(9,21); X_{11}(10,22); X_{12}(11,23); X_{13}(12,24); X_{14}(25,37); X_{15}(26,38); X_{16}(27,39); X_{17}(28,40);$ $X_{18}(29,41); X_{19}(30,42); X_{20}(31,43); X_{21}(32,44); X_{22}(33,45); X_{23}(34,46); X_{24}(35,47); X_{25}(36,48)$ |
| Size variables (in$^2$) | [0.5, 0.6, 0.7, . . ., 4.9] |

results obtained from MOHO were compared with those from MOAS, MOACS, DEMO, NSGA-2, MOALO, and MOMFO.

- The hyper-volume (HV) index measures the percentage of target space that members of the ND solution occupy set S. It provides information about the S set's convergence and diversity. A hypercube vi is created for every solution i in S by a collection of reference points. A higher HV value denotes an algorithm that performs better. A visual representation of HV for MO problems is shown in Fig 8.

$$HV = volume(\overset{A}{\underset{i=1}{U}} V_i)$$

- The gap between the actual Pareto-optimal front and the estimated Pareto-optimal front found during the search process is measured by the Generational Distance (GD). The actual and approximate Pareto-optimal fronts coincide when the GD value is zero.

$$GD = \frac{\sqrt{\sum_{i=1}^{no} d_i^2}}{|P|}$$

**Table 7. Design configuration of the 72-bar problem.**

| The 72-bar Truss Structures | |
|---|---|
| Design variables | $X_i, i = 1,2,\ldots 6$ |
| Constraints ($\sigma^{max}$ in ksi) | 25 |
| Density ($\rho$ in lb/in$^3$) | 0.1 |
| Young modules (E in ksi) | 10000 |
| loading conditions (kips) | Case 1: $F_{1x} = F_{1y} = 5, F_{1z} = -5$ |
| | Case 2: $F_{1z} = F_{2z} = F_{3z} = F_{4z} = -5$ |
| Grouping of Members | $X_1(1–4); X_2(5–12); X_3(13–16); X_4(17,18); X_5(19–22); X_6(23–30); X_7(31–34); X_8(35,36);$ $X_9(37–40); X_{10}(41–48); X_{11}(49–52); X_{12}(53,54); X_{13}(55–58); X_{14}(59–66); X_{15}(67–70);$ $X_{16}(71,72)$ |
| Size variables (in$^2$) | [0.1, 0.2, 0.3, . . ., 2.5] |

**Table 8. Design configuration of 942-bar problem.**

| The 942-bar truss Structures | |
|---|---|
| Design variables | $X_i$, $i = 1,2,...,59$ |
| Constraints ($\sigma^{max}$ in ksi) | 25 |
| Density ($\rho$ in lb/in$^3$) | 0.1 |
| Young modules (E in ksi) | 10000 |
| loading conditions (kips) | Nodal conditions: |
| | Downward forces: |
| | Portion-1;$P_z = -3$ |
| | Portion-2; Pz = -6 |
| | Portion-3; Pz = -9 |
| | The loading of laterally: |
| | Right side; $P_x = 1.5$ |
| | Left side; $P_x = 1.0$ |
| | Lateral Loading: $P_y = 1.0$ |
| Grouping of Members | $X_1(1,2)$; $X_2(3–10)$; $X_3(11–18)$; $X_4(19–34)$; $X_5(35–46)$; $X_6(47–58)$; $X_7(59–82)$; $X_8(83–86)$; $X_9(87–90)$; $X_{10}(97–98)$; $X_{11}$ (99–106); $X_{12}(107–122)$; $X_{13}(123–130)$; $X_{14}(131–162)$; $X_{15}(163–170)$; $X_{16}(171–186)$; $X_{17}$ (187–194); $X_{18}(195–226)$; $X_{19}(227–234)$; $X_{20}(235–258)$; $X_{21}(259–270)$; $X_{22}(271–318)$; $X_{23}(319–330)$; $X_{24}$ (331–338); $X_{25}(339–342)$; $X_{26}(343–350)$; $X_{27}(351–358)$; $X_{28}(359–366)$; $X_{29}(367–382)$; $X_{30}(383–390)$; $X_{31}$ (391–398); $X_{32}(399–430)$; $X_{33}(431–446)$; $X_{34}(447–462)$; $X_{35}(463–486)$; $X_{36}$ (487–498); $X_{37}(499–510)$; $X_{38}(511–558)$; $X_{39}(559–582)$; $X_{40}(583–606)$; $X_{41}$ (607–630); $X_{42}(631–642)$; $X_{43}(643–654)$; $X_{44}(655–702)$; $X_{45}(703–726)$; $X_{46}$ (727–750); $X_{47}(751–774)$; $X_{48}(775–786)$; $X_{49}(787–798)$; $X_{50}(799–846)$; $X_{51}(847–870)$; $X_{52}(871–894)$; $X_{53}$ (895–902); $X_{54}(903–906)$; $X_{55}(907–910)$; $X_{56}(911–918)$; $X_{57}(919–926)$; $X_{58}(927–934)$; $X_{59}(935–942)$ |
| Size variables (in$^2$) | [1, 2, 3, ..., 200] |

- The distance between each reference point and the resulting Pareto-optimal front is compared and evaluated using the Inverted Generational Difference (IGD). A closer distance between the obtained front and the reference locations is indicated by lower values of both GD and IGD, which suggests that the algorithm performed well.

$$IGD = \frac{\sqrt{\sum_{i=1}^{nt}(d_i'^2)}}{|P'|}$$

In the given equation, |P| denotes the count of outcomes in the Pareto front, where *di* represents the Euclidean distance across the nearest solution from the reference front and the objective function vector of the ith solution in the acquired front. Conversely, |P'| signifies the quantity of the solutions for the reference plane. This metric is utilized to measure both front expansions and progression. Pictorial representation of GD and IGD matrices are shown in Fig 9 which assesses the proximity of solutions in the approximation set to the true Pareto front.

- Together, the metrics for extent (ET) and spacing (SP) create a new evaluation matrix called the spacing to extent (STE) ratio. This matrix allows one to examine the extent and

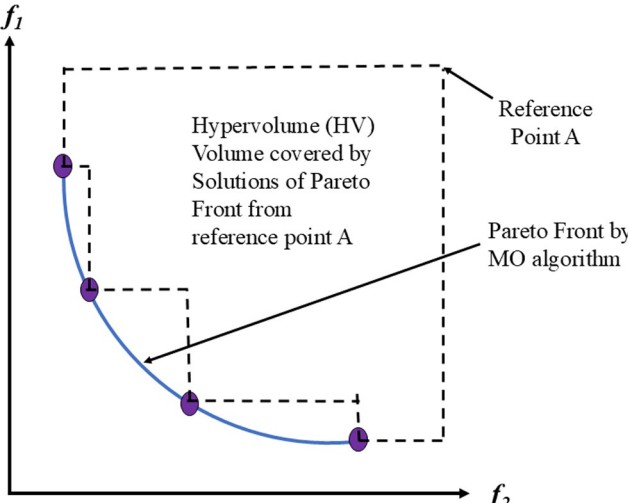

**Fig 8. Schematic view of HV matric.**

spacing aspects simultaneously. A more efficient and non-dominated Pareto front usually has a lower STE value.

$$SP = \frac{1}{|P| - 1} \sum_{i=1}^{|P|} (d_i - \bar{d})^2$$

$$ET = \sum_{i=1}^{M} |f_i^{\max} - f_i^{\min}|$$

The Euclidean distance $di$ separates the objective function vector of the i[th] solution from its closest neighbor. $\bar{d}$ is the mean value of all $di$, where M is the number of objective functions. $f_i^{\max}$ and $f_i^{\min}$ represent the maximum and minimum values of the i[th] objective

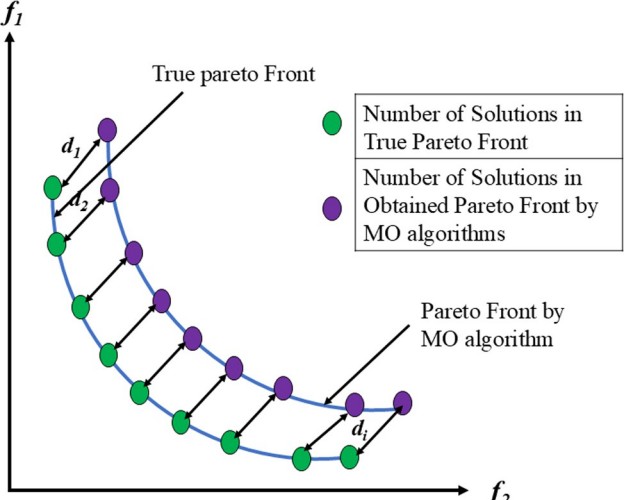

**Fig 9. Schematic view of GD and IGD matric.**

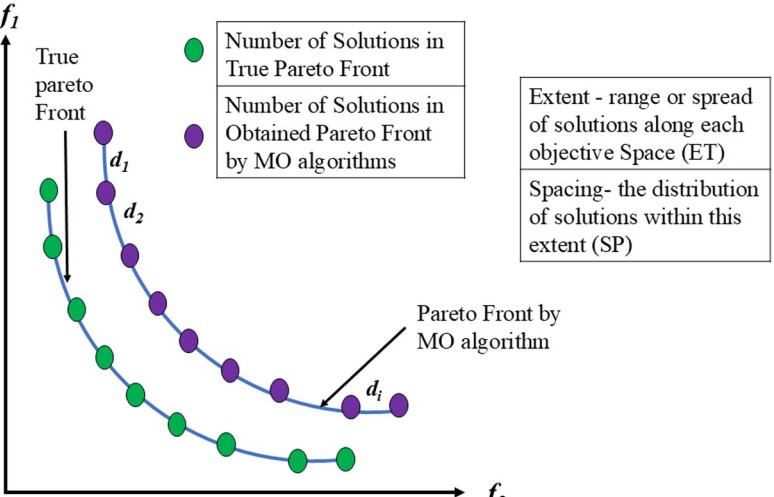

**Fig 10. Schematic view of SP and ET matric.**

function of the front, respectively. Fig 10 represents Spacing and Extent which quantifies how well solutions explore the objective space, providing insights into the diversity and coverage of the solution set.

## 5. Results-discussions and comparative study

The outcomes of the algorithms that are being examined are shown in below Tables 9–12, which are elucidated as follows:

Table 9 shows the hypervolume (HV) measures, which indicate how the algorithms' ND sorting capability has evolved and changed over time. Greater HV values are associated with superior non-dominated fronts. Out of all the problems examined, MOHO had the highest HV values, followed by MOMFO and DEMO, which suggests that it performed better in reaching a reasonable convergence rate.

- Variation in Hypervolume (HV): For all five considered truss constructions, MOHO shows significantly better average, maximum, and minimum HV values than MOMFO, MOALO, DEMO, NSGA-2, MOACS, and MOAS. Furthermore, MOHO's standard (std) value from the HV test is excellent for all the algorithms considered for all truss constructions, indicating better search consistency.

- Performance Comparison: Among all the average HV values of the algorithms considered, MOHO exhibits the highest values for truss structures, with respective averages of 53518.79, 1894.07, 4087.08, 2317.36, and 75288799 for 10-bar, 25-bar, 60-bar, 72-bar and 942-bar truss respectively. Algorithms with higher average HV values are more effective at exploring the solution space and identifying Pareto-optimal solutions.

- MO algorithm effectiveness: Comparing the HV values for each MO algorithm across the different truss structures allows us to identify which algorithms consistently perform well across various optimization problems.

- Standard deviation analysis: Std value by the MOHO for all considered trusses are less compared with other optimizers with Friedman values of 1.10, 1, 1, 1.03, and 1 for 10-bar, 25-bar, 60-bar, 72-bar and 942-bar truss respectively. The stability and consistency of

**Table 9. The hypervolume (HV).**

| HV | | MOAS | MOACS | DEMO | NSGA-2 | MOALO | MOMFO | MOHO |
|---|---|---|---|---|---|---|---|---|
| 10-bar | average | 39544.95 | 36065.71 | 51947.63 | 35074.27 | 38218.11 | 52640.18 | **53518.79** |
| | max | 51605.99 | 44798.06 | 53198.54 | 40975.08 | 48996.76 | 53408.43 | **54317.72** |
| | min | 8251.52 | 28933.63 | 48747.70 | 0.00 | 23998.66 | 51405.56 | **52753.20** |
| | std | 9651.11 | 3752.74 | 923.17 | 7374.47 | 5759.18 | 475.15 | **456.45** |
| | Friedman | 4.97 | 5.77 | 2.80 | 5.90 | 5.37 | 2.10 | **1.10** |
| 25-bar | average | 1027.19 | 1628.28 | 1639.65 | 1601.52 | 1374.24 | 1858.88 | **1894.07** |
| | max | 1653.51 | 1761.40 | 1840.80 | 1676.04 | 1764.46 | 1881.47 | **1906.93** |
| | min | 121.93 | 1512.45 | 695.96 | 1469.97 | 901.02 | 1791.72 | **1872.52** |
| | std | 497.46 | 68.94 | 267.85 | 43.66 | 255.23 | 18.05 | **9.28** |
| | Friedman | 6.57 | 4.20 | 3.87 | 4.67 | 5.70 | 2.00 | **1.00** |
| 60-bar | average | 3188.41 | 3301.08 | 3624.14 | 2691.17 | 3219.26 | 3977.46 | **4087.08** |
| | max | 4096.04 | 3940.78 | 3840.72 | 2902.64 | 3681.27 | 4019.09 | **4108.55** |
| | min | 84.85 | 2610.09 | 3389.52 | 2424.62 | 2467.17 | 3913.57 | **4056.28** |
| | std | 1189.35 | 544.85 | 114.15 | 100.83 | 327.38 | 30.00 | **11.93** |
| | Friedman | 4.20 | 4.73 | 4.17 | 6.50 | 5.17 | 2.23 | **1.00** |
| 72-bar | average | 1931.19 | 1647.99 | 2104.08 | 1645.71 | 2013.06 | 2275.27 | **2317.36** |
| | max | 2218.42 | 1875.97 | 2269.07 | 1765.59 | 2196.28 | 2305.10 | **2327.06** |
| | min | 1212.83 | 1513.92 | 1915.60 | 1559.00 | 1742.23 | 2209.54 | **2297.54** |
| | std | 288.28 | 71.42 | 80.71 | 55.37 | 139.62 | 18.29 | **8.62** |
| | Friedman | 4.53 | 6.30 | 3.63 | 6.33 | 4.20 | 1.97 | **1.03** |
| 942-bar | average | 56123236 | 47412983 | 61052456 | 47050336 | 65884888 | 67704395 | **75288799** |
| | max | 61351328 | 50136472 | 65153422 | 49836195 | 69685856 | 69643419 | **76194683** |
| | min | 49584839 | 45253375 | 55362485 | 44692631 | 60607421 | 65296967 | **72369750** |
| | std | 2941688 | 1272304 | 2314905 | 1437399 | 1914314 | 1327312 | **787741** |
| | Friedman | 4.83 | 6.40 | 4.17 | 6.60 | 2.70 | 2.30 | **1.00** |
| Average Friedman | | 5.02 | 5.48 | 3.73 | 6.00 | 4.63 | 2.12 | **1.03** |
| Overall Friedman rank | | 5 | 6 | 3 | 7 | 4 | 2 | **1** |

algorithm performance can be inferred from the standard deviation of HV values. While higher values might suggest variability or sensitivity to specific problem instances, lower standard deviation values indicate more consistent performance.

- Sensitivity of truss structures: The MOHO framework balances exploitation, prioritizing feasible solutions, and exploration, which involves seeking diverse solution options, and offering a well-rounded optimization approach. With these capabilities, MOHO holds promise in systematically navigating the vast solution space of larger structures, potentially approaching optimal designs.

Rigorous comparisons using Friedman's rank test rank algorithms, with MOHO emerging with the top score of 1.03 among all algorithms with conflicting objectives, indicating superior Pareto front quality compared to alternatives. Analyzing the HV values in the table enables a comprehensive assessment of multi-objective optimisation algorithms' effectiveness, robustness, and sensitivity across a range of truss structures, facilitating informed decision-making in algorithm selection and application.

Figs 11–15 depict the Pareto fronts for truss problems across seven algorithms. They reveal the correlation between mass and maximum displacement, which provides visual insights into the trade-offs between these objectives. Each figure represents a scatter plot showing the

**Table 10. The GD measures of the considered truss structures.**

| GD | | MOAS | MOACS | DEMO | NSGA-2 | MOALO | MOMFO | MOHO |
|---|---|---|---|---|---|---|---|---|
| 10-bar | average | 5.6994 | 2.9652 | 4.7677 | 3.33E+08 | 1.9398 | 3.8949 | 4.1908 |
| | max | 8.0166 | 6.8208 | 5.8473 | 1.00E+10 | 4.9412 | 7.0546 | 5.1374 |
| | min | 2.5772 | 0.4452 | 2.5616 | 8.9381 | 0.8002 | 1.9263 | 2.7504 |
| | std | 1.3346 | 1.6123 | 0.7484 | 1.83E+09 | 0.9160 | 1.1929 | 0.5805 |
| | Friedman | 5.4000 | 2.5333 | 4.2333 | 7.0000 | **1.3000** | 3.5000 | 4.0333 |
| 25-bar | average | 0.5627 | 0.3446 | 0.6708 | 1.2910 | 0.0832 | 0.3477 | 0.4071 |
| | max | 0.7719 | 0.7302 | 2.0403 | 2.0228 | 0.1867 | 0.5953 | 1.2124 |
| | min | 0.2822 | 0.1266 | 0.2601 | 0.6063 | 0.0346 | 0.1102 | 0.2720 |
| | std | 0.1262 | 0.1236 | 0.4695 | 0.2803 | 0.0364 | 0.1011 | 0.1666 |
| | Friedman | 5.2333 | 2.9667 | 4.9667 | 6.9000 | **1.0000** | 3.2333 | 3.7000 |
| 60-bar | average | 0.3143 | 0.3324 | 0.6781 | 0.2983 | 0.1286 | 0.2941 | 0.6826 |
| | max | 0.7472 | 1.9680 | 2.1178 | 1.1024 | 0.2075 | 0.5852 | 1.4381 |
| | min | 0.0000 | 0.1343 | 0.3318 | 0.0988 | 0.0591 | 0.1795 | 0.2847 |
| | std | 0.1912 | 0.3232 | 0.4397 | 0.1912 | 0.0468 | 0.0896 | 0.3439 |
| | Friedman | 3.6333 | 3.5000 | 6.2000 | 3.4333 | **1.4667** | 3.6333 | 6.1333 |
| 72-bar | average | 1.0229 | 0.9811 | 3.5130 | 3.1078 | 0.2492 | 0.6245 | 1.4976 |
| | max | 1.2480 | 1.9517 | 8.2646 | 4.7378 | 0.6301 | 1.0736 | 4.4378 |
| | min | 0.6030 | 0.2402 | 1.0332 | 1.9911 | 0.0982 | 0.2532 | 0.5908 |
| | std | 0.1489 | 0.4964 | 1.8632 | 0.6718 | 0.1110 | 0.2253 | 1.0042 |
| | Friedman | 4.0000 | 3.5000 | 6.3000 | 6.4000 | **1.1000** | 2.4000 | 4.3000 |
| 942-bar | average | 2667.60 | 2134.09 | 1912.15 | 8970.45 | 1266.71 | 1936.82 | 1609.70 |
| | max | 3106.67 | 5640.03 | 2203.94 | 13726.12 | 2080.17 | 3584.51 | 2028.71 |
| | min | 2174.14 | 244.77 | 1649.89 | 4312.12 | 594.68 | 1287.83 | 1133.31 |
| | std | 251.52 | 1209.88 | 159.86 | 2461.80 | 368.18 | 509.06 | 230.60 |
| | Friedman | 5.57 | 3.80 | 3.67 | 7.00 | **1.67** | 3.57 | 2.73 |
| Average Friedman | | 4.77 | 3.26 | 5.07 | 6.15 | 1.31 | 3.27 | 4.18 |
| Overall Friedman rank | | 5 | 2 | 6 | 7 | 1 | 3 | 4 |

distribution of solutions along the Pareto front, where each point represents a unique solution generated by the optimization algorithm.

• Truss Structures: The figures display Pareto fronts for different truss structures, such as 10-bar, 25-bar, 60-bar, 72-bar, and 942-bar, allowing for comparisons across various design complexities and sizes.

• MO Optimization Algorithms: The plots include Pareto fronts generated by seven multi-objective optimization algorithms, such as MOAS, MOACS, DEMO, NSGA-2, MOALO, MOMFO, and MOHO. Each algorithm employs distinct optimization strategies and techniques to explore the solution space and identify Pareto-optimal solutions.

• Mass vs. Displacement: The scatter plots' axes represent the two objectives of interest: mass (structural weight) and displacement (structural deflection). The position of each point on the plot indicates the corresponding values of mass and displacement for a particular design solution.

• Trade-Off Analysis: The researcher can analyse the trade-offs between both objectives by examining the distribution of points along the Pareto front. Solutions located closer to the Pareto front represent superior trade-offs, where improvements in one objective (e.g., reducing mass) come at the expense of the other (e.g., increasing displacement).

**Table 11. The IGD measures of the considered truss structures.**

| IGD | | MOAS | MOACS | DEMO | NSGA-2 | MOALO | MOMFO | MOHO |
|---|---|---|---|---|---|---|---|---|
| 10-bar | average | 245.5037 | 259.1469 | 24.5485 | 3.33E+08 | 337.7231 | 45.6893 | **17.8805** |
| | max | 689.9251 | 363.2606 | 113.5396 | 1.00E+10 | 428.7546 | 101.0405 | **57.6933** |
| | min | 93.1421 | 174.9454 | 9.4067 | 185.3922 | 105.2070 | 27.1297 | **8.8335** |
| | std | 131.8729 | 42.6827 | 23.4657 | 1.83E+09 | 71.3010 | 16.8154 | **9.1157** |
| | Friedman | 4.9000 | 5.2333 | 1.7333 | 5.3333 | 6.5333 | 2.9333 | **1.3333** |
| 25-bar | average | 34.3229 | 11.0436 | 4.6787 | 12.0720 | 31.0667 | 3.4426 | **1.2505** |
| | max | 72.5876 | 17.5043 | 15.9252 | 16.1651 | 39.1427 | 6.3338 | **2.5849** |
| | min | 7.5076 | 5.4122 | 0.8751 | 8.5116 | 6.8781 | 2.1939 | **0.6056** |
| | std | 21.1155 | 3.1577 | 3.7489 | 1.9336 | 7.4651 | 0.9523 | **0.4541** |
| | Friedman | 6.2667 | 4.4333 | 2.5333 | 4.7333 | 6.4667 | 2.4667 | **1.1000** |
| 60-bar | average | 15.1235 | 6.8565 | 3.6174 | 9.1537 | 21.8567 | 3.3711 | **0.7495** |
| | max | 61.7861 | 10.7323 | 6.9676 | 10.8502 | 30.5357 | 6.6360 | **1.1366** |
| | min | 0.6520 | 0.4888 | 1.6015 | 7.3412 | 6.1109 | 2.0186 | **0.4451** |
| | std | 17.5571 | 2.7051 | 1.4009 | 0.9089 | 5.8313 | 1.0752 | **0.1851** |
| | Friedman | 4.8333 | 4.2333 | 3.0667 | 5.2000 | 6.7333 | 2.8667 | **1.0667** |
| 72-bar | average | 27.1851 | 38.8656 | 9.0814 | 39.0035 | 41.9450 | 7.5117 | **1.6302** |
| | max | 70.3734 | 47.9222 | 19.8944 | 45.5011 | 60.0562 | 17.9598 | **2.8637** |
| | min | 8.1065 | 25.4738 | 2.2300 | 30.2875 | 12.1682 | 4.5928 | **0.9054** |
| | std | 17.5471 | 4.9933 | 4.0344 | 4.3014 | 14.0874 | 2.8257 | **0.5768** |
| | Friedman | 4.6000 | 5.7000 | 2.6667 | 5.6333 | 6.0000 | 2.4000 | **1.0000** |
| 942-bar | average | 76437.04 | 98051.79 | 44304.17 | 98205.29 | 72188.48 | 39564.85 | **7718.49** |
| | max | 92320.10 | 110780.67 | 66006.99 | 107838.17 | 118683.48 | 63580.56 | **23055.91** |
| | min | 59161.51 | 82630.70 | 31057.57 | 89416.37 | 26295.28 | 28226.57 | **2355.06** |
| | std | 10315.08 | 5733.64 | 7262.66 | 5003.67 | 24157.40 | 6832.37 | **6110.29** |
| | Friedman | 4.6000 | 6.2667 | 2.7333 | 6.4000 | 4.5000 | 2.5000 | **1.0000** |
| Average Friedman | | 5.04 | 5.17 | 2.55 | 5.46 | 6.05 | 2.63 | **1.10** |
| Overall Friedman rank | | 4 | 5 | 2 | 6 | 7 | 3 | **1** |

- Algorithm Performance: The distribution and spread of points along the Pareto front provide insights into the performance of each optimization algorithm. Algorithms that generate solutions distributed across a wide range of Pareto front regions demonstrate better exploration capabilities and versatility in identifying diverse trade-off solutions.

This observation underscores MOHO's superior performance. Its Pareto fronts are characterized by smoothness and even distribution, in stark contrast to the fragmented and discontinuous fronts generated by other multi-objective algorithms. This distinction highlights MOHO's prowess in optimizing truss structures more efficiently and effectively across various design objectives.

The hypervolume generation process with the function evaluation of different truss constructions is seen in Figs 16–20. The hypervolume metric's evolution during the optimization process is depicted graphically in these figures, which also highlight the various algorithms' abilities to explore and converge towards the Pareto front. By scrutinizing these figures, it becomes apparent how effectively each algorithm performs concerning coverage and convergence toward optimal trade-off solutions. Also, the figures illustrating hypervolume through function evaluations for all MO optimization algorithms across various truss structures provide insights into the performance of each algorithm over time. By plotting the hypervolume

**Table 12. The STE measures for the all-truss problems.**

| STE | | MOAS | MOACS | DEMO | NSGA-2 | MOALO | MOMFO | MOHO |
|---|---|---|---|---|---|---|---|---|
| 10-bar | average | 0.0297 | 0.0632 | 0.0063 | 3.33E+18 | 0.0337 | 0.0058 | 0.0110 |
| | max | 0.0849 | 0.1000 | 0.0137 | 1.00E+20 | 0.0655 | 0.0195 | 0.0177 |
| | min | 0.0026 | 0.0108 | 0.0039 | 0.0064 | 0.0000 | 0.0000 | 0.0054 |
| | std | 0.0199 | 0.0286 | 0.0019 | 1.83E+19 | 0.0193 | 0.0052 | 0.0032 |
| | Friedman | 4.5333 | 5.8000 | **1.7667** | 6.5667 | 4.4333 | 1.8333 | 3.0667 |
| 25-bar | average | 0.0286 | 0.0228 | 0.0065 | 0.0565 | 0.0304 | 0.0072 | 0.0110 |
| | max | 0.0846 | 0.0430 | 0.0162 | 0.1156 | 0.0531 | 0.0207 | 0.0171 |
| | min | 0.0000 | 0.0085 | 0.0020 | 0.0230 | 0.0058 | 0.0000 | 0.0056 |
| | std | 0.0230 | 0.0070 | 0.0028 | 0.0259 | 0.0187 | 0.0061 | 0.0026 |
| | Friedman | 4.6167 | 4.8333 | **1.9667** | 6.6000 | 4.6333 | 2.2500 | 3.1000 |
| 60-bar | average | 0.0252 | 0.0114 | 0.0069 | 0.0194 | 0.0345 | 0.0063 | 0.0077 |
| | max | 0.0995 | 0.0214 | 0.0193 | 0.0368 | 0.0515 | 0.0211 | 0.0121 |
| | min | 0.0000 | 0.0059 | 0.0036 | 0.0065 | 0.0015 | 0.0000 | 0.0046 |
| | std | 0.0290 | 0.0041 | 0.0032 | 0.0081 | 0.0161 | 0.0061 | 0.0016 |
| | Friedman | 4.7333 | 4.2333 | **2.1667** | 5.4000 | 6.1000 | 2.5333 | 2.8333 |
| 72-bar | average | 0.0162 | 0.0285 | 0.0058 | 0.0557 | 0.0328 | 0.0056 | 0.0089 |
| | max | 0.0382 | 0.0450 | 0.0086 | 0.1051 | 0.0520 | 0.0187 | 0.0128 |
| | min | 0.0069 | 0.0171 | 0.0042 | 0.0187 | 0.0000 | 0.0000 | 0.0058 |
| | std | 0.0081 | 0.0072 | 0.0014 | 0.0185 | 0.0192 | 0.0061 | 0.0017 |
| | Friedman | 4.2667 | 5.2000 | **1.7667** | 6.7667 | 5.0333 | 2.1000 | 2.8667 |
| 942-bar | average | 0.0125 | 0.0366 | 0.0073 | 0.1135 | 0.0364 | 0.0096 | 0.0088 |
| | max | 0.0238 | 0.0651 | 0.0183 | 0.4678 | 0.0710 | 0.0185 | 0.0129 |
| | min | 0.0060 | 0.0124 | 0.0015 | 0.0240 | 0.0007 | 0.0010 | 0.0059 |
| | std | 0.0044 | 0.0143 | 0.0035 | 0.0812 | 0.0162 | 0.0044 | 0.0017 |
| | Friedman | 3.4667 | 5.5000 | **1.7667** | 6.8667 | 5.2333 | 2.8667 | 2.3000 |
| Average Friedman | | 4.3233 | 5.1133 | **1.8867** | 6.4400 | 5.0867 | 2.3167 | 2.8333 |
| Overall Friedman rank | | 4 | 6 | **1** | 7 | 5 | 2 | 3 |

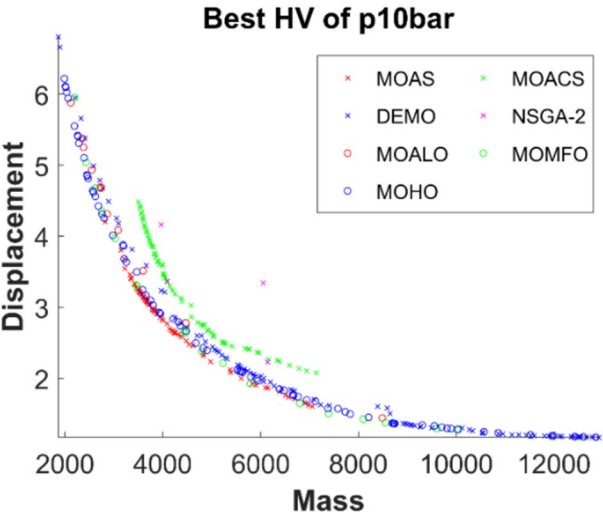

**Fig 11. The finest Pareto fronts of the 10-bar truss.**

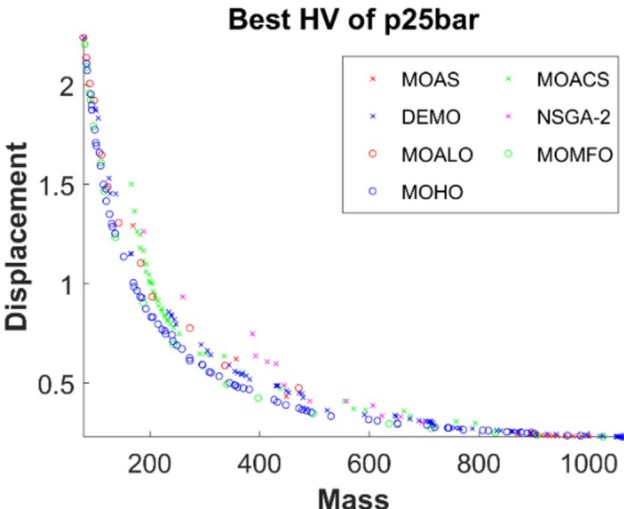

**Fig 12. The finest Pareto fronts of the 25-bar truss.**

values against the number of function evaluations, these figures showcase how efficiently each algorithm converges towards the Pareto-optimal front. A steeper increase in hypervolume indicates faster convergence towards a better Pareto front. Additionally, comparing the hypervolume curves of different algorithms allows for assessing their relative efficiency and effectiveness in exploring the solution space and identifying high-quality solutions. These figures serve as a visual representation of the optimization process's progress and enable researchers to evaluate the algorithm's performance dynamically. This empirical analysis helps to clarify how different optimization approaches compare in terms of their ability to solve multi-objective truss design problems.

The findings of the Generational Distance (GD) metric are shown in Table 10, which is an essential metric for evaluating the differences between the Pareto optimal front and ND solutions in various truss configurations. A lower GD score indicates an exceptional, non-dominated front. For the 10-bar truss average GD value is 4.1908 with a standard deviation (std) of

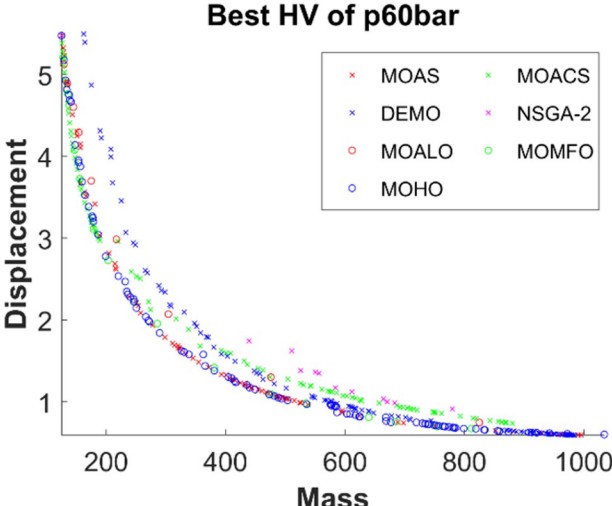

**Fig 13. The finest Pareto fronts of the 60-bar truss.**

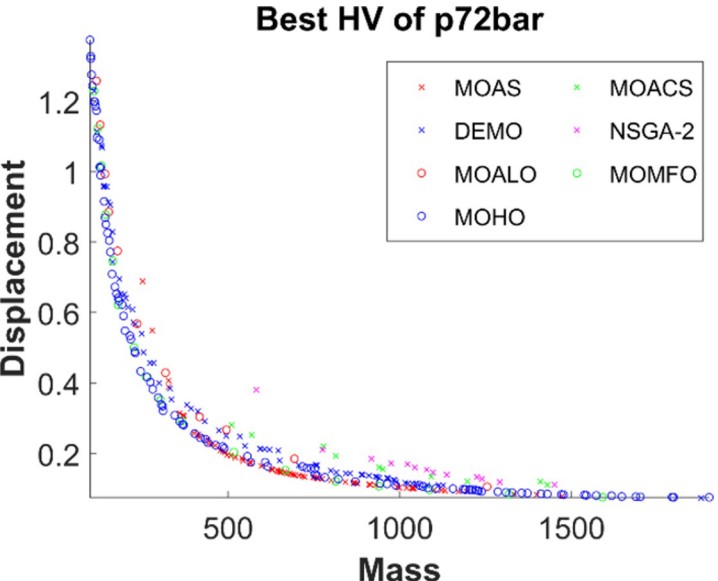

**Fig 14. The finest Pareto fronts of the 72-bar truss.**

0.5805, for the 25-bar it is 0.4071 with a std 0.1666, for the 60-bar it is 0.6826 with a std of 0.3439, for the 72-bar it is 1.4976 with a std of 1.0042 and for the 942-bar truss structure average GD is 1609.70 with a std of 230.60 for MOHO. The GD measure results reveal that MOALO, MOACS, MOMFO, and MOHO have excellent non-dominated fronts and perform extraordinarily well, which means they can generate solutions closer to the true Pareto front and well-distributed across the objective space. On the other hand, MOAS, DEMO, and NSGA-2 perform worse when evaluated using the GD metric. These comparative analyses help to understand the performance of all considered MO algorithms for all considered five truss structures in terms of their ability to explore the solution space effectively and produce high-quality Pareto fronts.

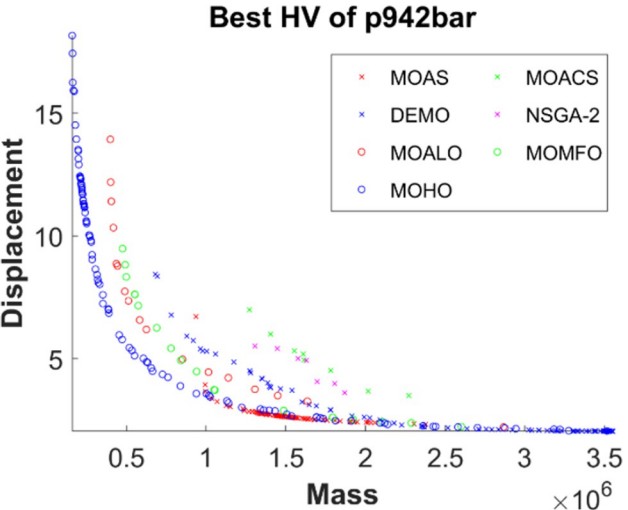

**Fig 15. The finest Pareto fronts of the 942-bar truss.**

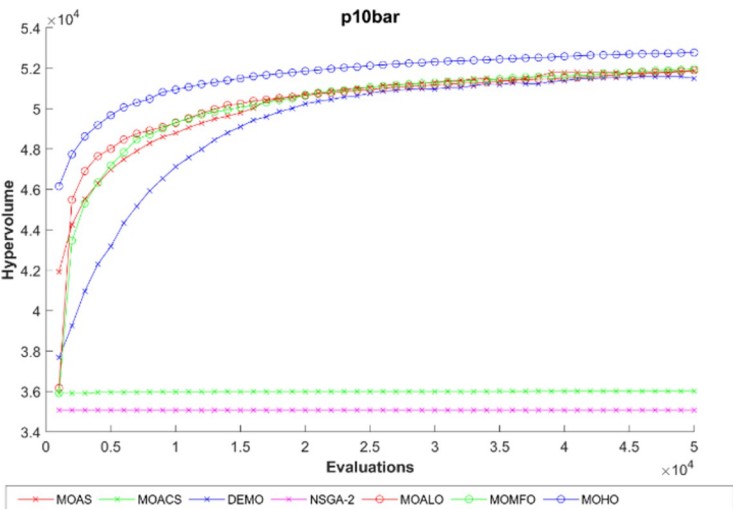

**Fig 16. The hypervolume through function evaluations of the 10-bar truss.**

A lower value of the Inverted Generational Distance (IGD) metric shows an improved ND front, which provides an all-encompassing assessment of convergence and diversity among Pareto fronts. The IGD measurements' results are shown in Table 11, which includes information on how different algorithms perform when compared to various truss structures. Regarding IGD values, MOHO leads to the front, followed by DEMO and MOMFO, all exhibiting better convergence and diversity in their Pareto fronts. The mean Euclidean distance between every point in the obtained front and the closest point in the actual Pareto front is measured. It assesses how close an algorithm's solutions are to the true Pareto front. Additionally, MOHO is at the top, with a first rank at a 95% significant level in Friedman's test, highlighting their efficacy in producing high-quality Pareto fronts. These results shed light on the relative merits and drawbacks of different methods in achieving convergence and diversity in MO optimization for truss design. They offer insightful information about the optimization procedure,

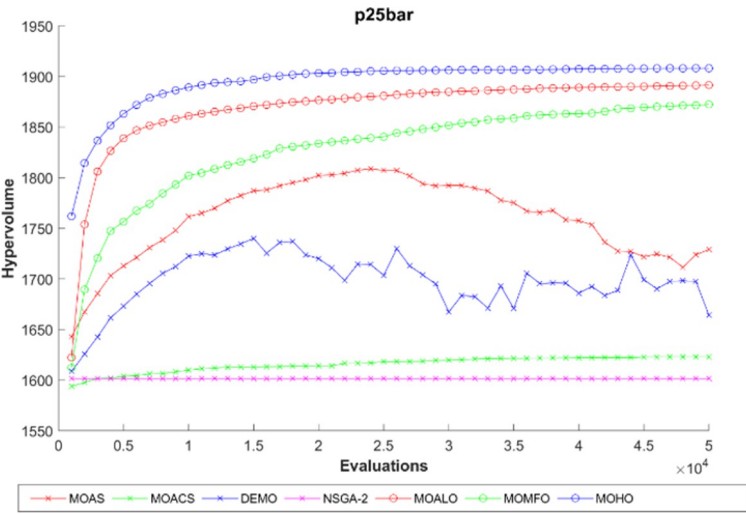

**Fig 17. The hypervolume through function evaluations of the 25-bar truss.**

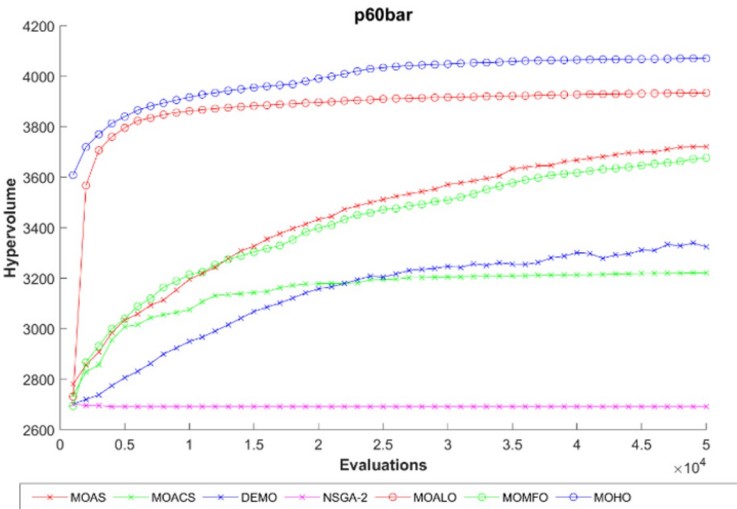

**Fig 18. The hypervolume through function evaluations of the 60-bar truss.**

which helps to clarify how various algorithms function when dealing with truss design difficulties.

Diversity curves illustrating the evolution of the diversity of solutions concerning function evaluation for all algorithms used for the truss structures under consideration are shown in Figs 21–25. These curves illustrate each algorithm's performance in exploring the solution space and preventing early convergence to poor solutions. They offer essential insights into how each approach maintains diversity throughout optimization. A higher diversity indicates a broader range of solutions the algorithm explores, potentially leading to a more comprehensive solution space exploration. Fluctuations or plateaus in the diversity curve may indicate transitions between exploration and exploitation phases or convergence to suboptimal solutions. Overall, these curves provide valuable insights into the algorithm's ability to explore diverse solution alternatives and avoid premature convergence to local optima. When choosing algorithms for MO optimization problems in truss design, researchers can make well-

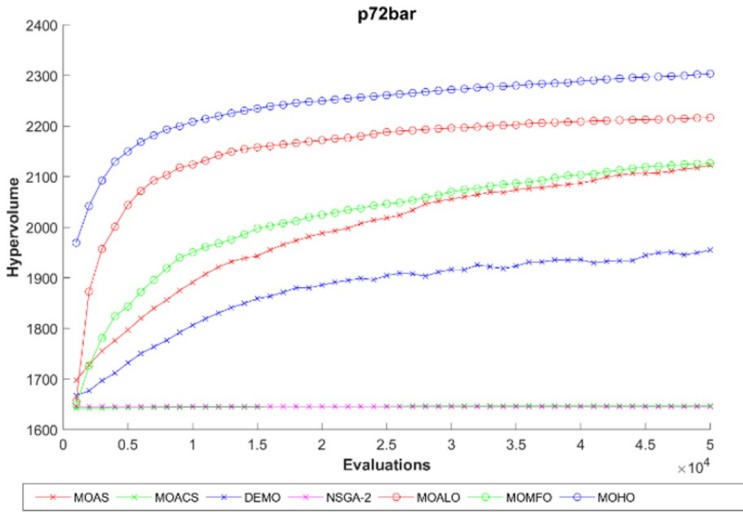

**Fig 19. The hypervolume through function evaluations of the 72-bar truss.**

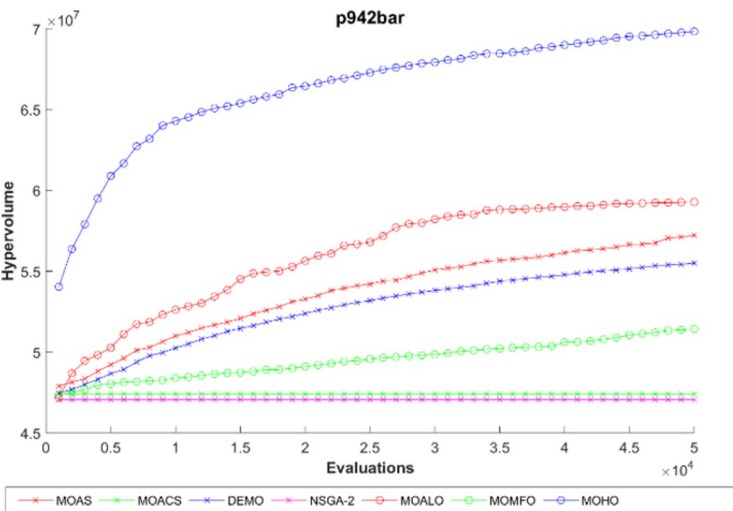

**Fig 20. The hypervolume through function evaluations of the 942-bar truss.**

informed decisions with the help of these figures, which are crucial tools for comprehending the dynamic behavior of optimization algorithms.

Spacing represents a good distribution of solutions with evenly spaced points, indicating that the MO algorithm has explored the objective space effectively and generated the most diverse solution set. Extent is a range of solutions along the objective space; a higher value means the solutions cover a wide range of objectives, indicating diversity. By simultaneously evaluating both spacing and extent, the Spacing-To-Extent (STE) metric offers essential information about the quality of non-dominated fronts. A lower STE score indicates a superior, ND front because it matches the spacing-to-extent ratio better. Table 12 presents the STE findings for the MO algorithms under consideration in the study. For 10-bar, 25-bar, and 72-bar trusses, the average STE value of MOHO is 0.0110, less than MOAS, MOACS, MOALO, and NSGA-2. For a 60-bar ring truss, an average STE value from MOHO is 0.0077 with the least std of 0.0089, the third best compared with other MO algorithms. Same for the large tower

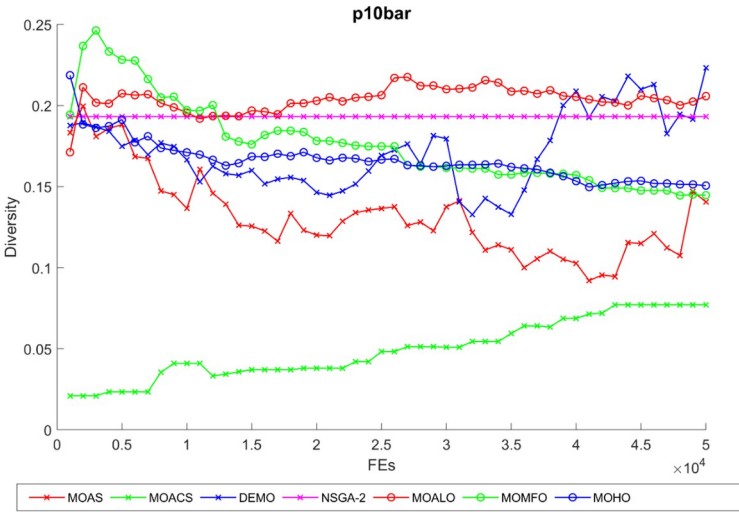

**Fig 21. The diversity curve of 10-bar truss problem.**

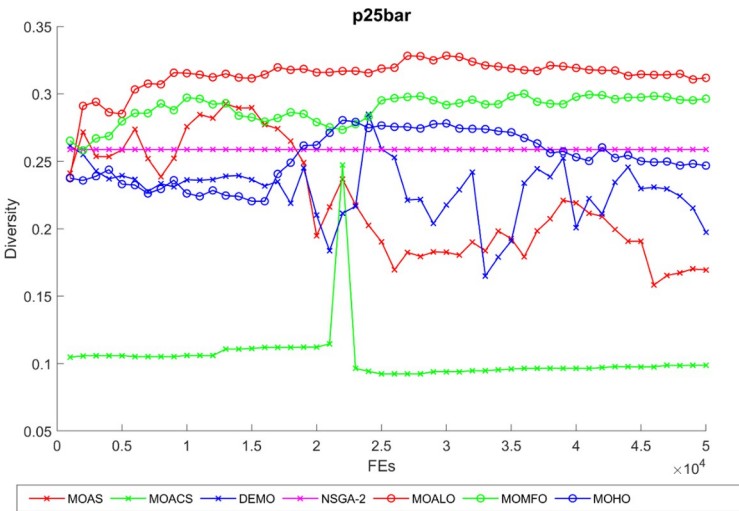

**Fig 22. The diversity curve of 25-bar truss problem.**

truss 942-bar, an average STE measure is 0.0088 with a std of 0.0017 with a Friedman value of 2.3. As can be seen from the STE findings, DEMO performs best, demonstrating its ability to create comprehensive, evenly spaced, non-dominated fronts. As the runner-up, MOMFO and MOHO exhibit STE results similar to DEMO's. According to Friedman's test at a 95% significance threshold, while other metrics may yield comparable findings, the overall Friedman test positions DEMO, MOMFO, and MOHO as the top three algorithms, reaffirming their effectiveness in generating high-quality non-dominated fronts with favourable spacing to extent ratios.

Figs 26–30 depict swarm plots representing the performance of optimization algorithms across all the truss structures, respectively. Swarm plots visually represent the distribution of objective function values or performance metrics obtained by different algorithms for each truss structure. These swarm plots provide information about how the solutions produced by each algorithm are distributed among the various truss structures. This analysis aids in

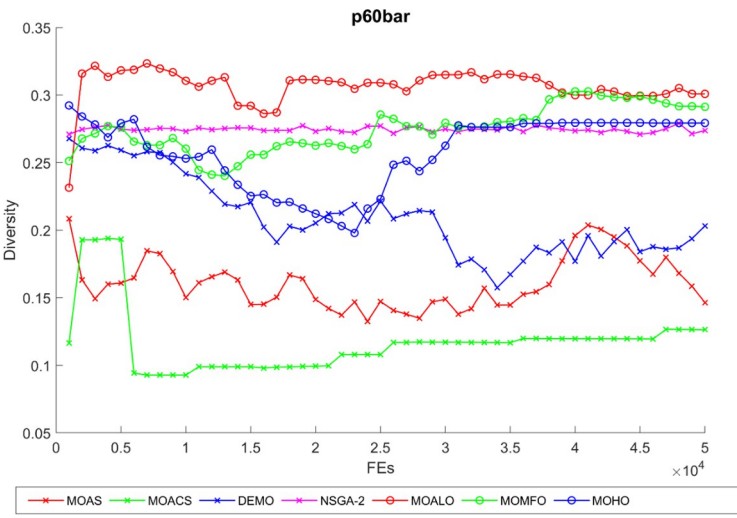

**Fig 23. The diversity curve of the 60-bar truss problem.**

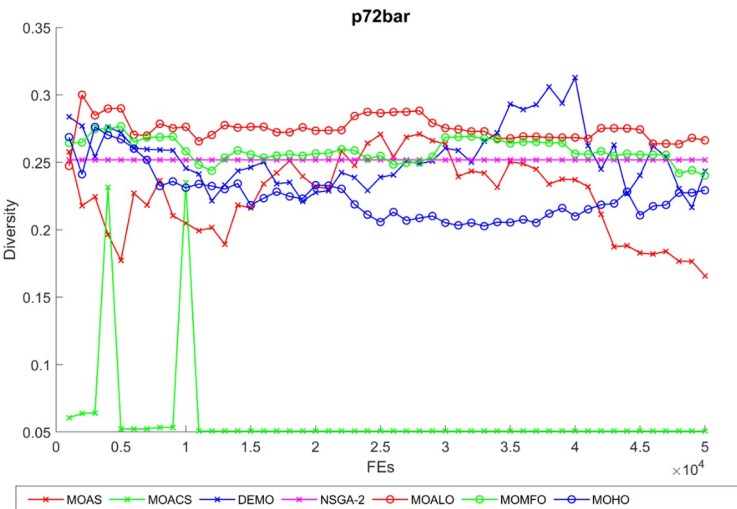

**Fig 24. The diversity curve of the 72-bar truss problem.**

identifying any potential outliers or patterns in the optimization process and understanding algorithm behaviour regarding solution quality and variety. These plots offer insights into how the population evolves, showing trends such as convergence towards some areas of the objective space or the distribution of solutions across different regions. By visualizing the swarm plots, researchers can observe the algorithm's ability to explore the solution space effectively and maintain diversity within the population. Variations in the density or dispersion of points in the swarm plot can indicate the algorithm's performance in balancing exploration and exploitation. Additionally, researchers can analyze how the swarm evolves in response to changes in algorithm parameters or problem settings. Overall, swarm plots provide a dynamic representation of the optimization process, allowing researchers to monitor and interpret the behaviour of MO optimization algorithms in the truss structure design. MOHO has a superior quality of swarm generation for all considered truss structures.

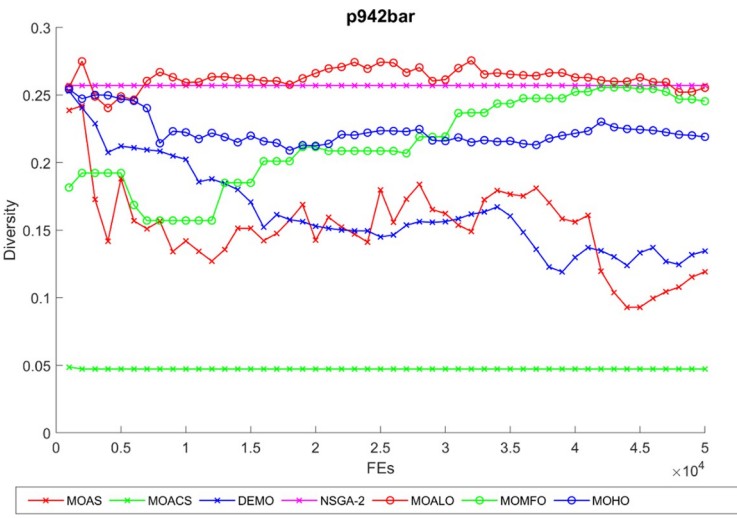

**Fig 25. The diversity curve of the 942-bar truss problem.**

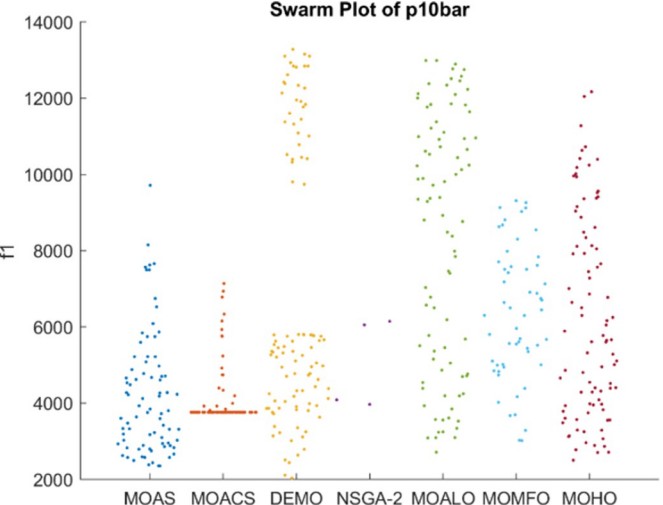

**Fig 26. Swarm plots of 10-bar truss.**

Box plots provide a visual summary of the distribution of objective function (HV) values or performance metrics obtained by different algorithms for each truss structure. These box plots evaluate the central tendency, variability, and dispersion of each method's solution quality for various truss structures. A higher median hypervolume value and a narrower IQR (interquartile range) indicate better overall performance and consistency of an algorithm in generating Pareto-optimal solutions. Conversely, a wider IQR and a more dispersed distribution of hypervolume values may suggest more significant variability or instability in the algorithm's performance. Box plots can be utilized to compare the overall potential distribution of different algorithms, spot any performance disparities, and learn more about how different algorithms behave when faced with varying levels of task complexity. Overall, Figs 31–35 are valuable tools that help assess and choose suitable algorithms for specific optimization problems by visually comparing the effectiveness of optimization algorithms over various truss structures.

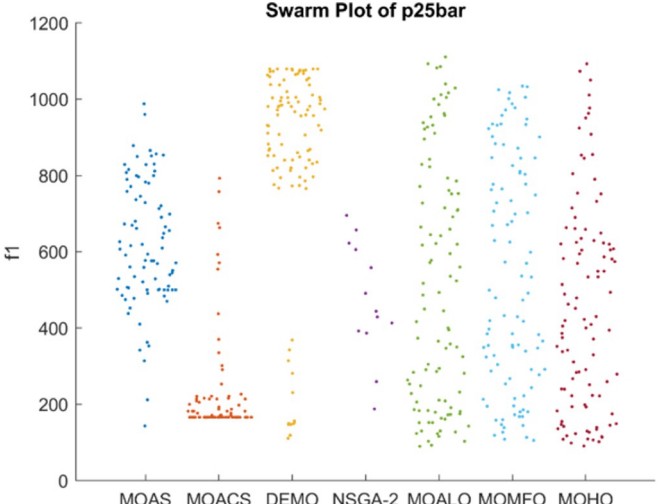

**Fig 27. Swarm plots of 25-bar truss.**

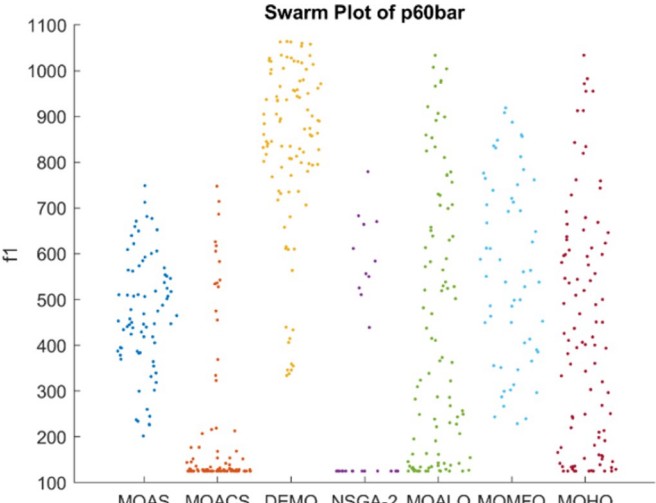

**Fig 28. Swarm plots of 60-bar truss.**

An overview of the comprehensive Friedman rank for all truss structures achieved by the techniques under consideration is shown in Table 13. The average Friedman's score for MOHO is 2.2850, which is the lowest and has the first rank compared with MOMFO, DEMO, MOALO, MOACS, MOAS, and NSGA-2. MOHO has an excellent convergence rate compared with the other prominent MO optimization algorithms. This dominance of MOHO is statistically significant, as indicated by Friedman's rank test at a 95% level, further underscoring its better performance than other algorithms assessed in the research. Overall, MOHO has the most excellent HV values, indicating that it explores well and comes up with various solutions. The results of MOHO, MOALO, MOACS, and MOMFO are near the optimal Pareto front, as indicated by their lowest GD values. In most instances, MOHO also had the lowest IGD, indicating a decent mix of convergence and diversity. MOHO was the most effective algorithm for these truss structure problems when ranked according to all three measures. Put more,

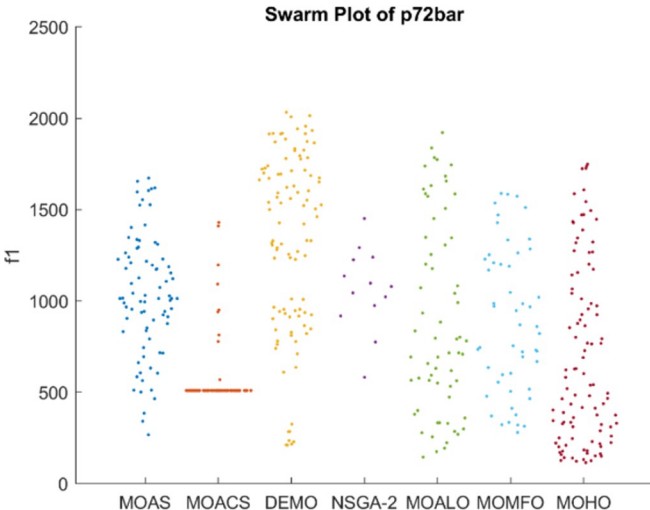

**Fig 29. Swarm plots of 72-bar truss.**

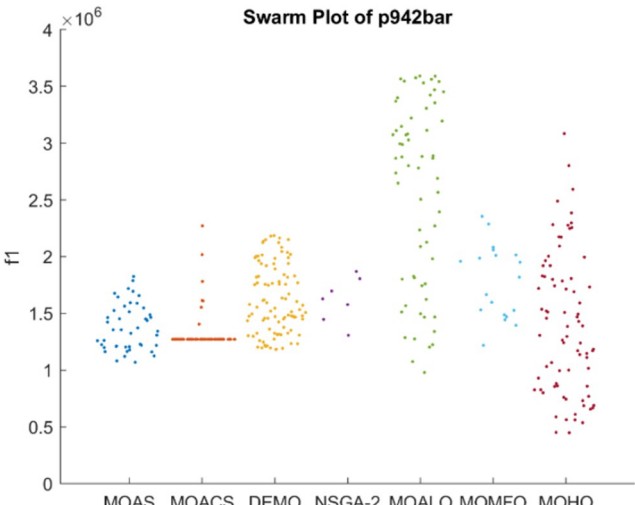

**Fig 30. Swarm plots of 942-bar truss.**

MOHO is the best preference for these challenging engineering investigations because it finds a good distribution of near-ideal, well-balanced solutions.

## 6. Conclusion

This article presents the MO version of the novel hippopotamus optimization algorithm for solving five structural truss problems. Minimization of both objective function, structural mass, and maximum nodal displacement is subject to stress and area constraints. This algorithm's two decisive exploration phases and one exploitation phase generate excellent results for the truss optimization problems to examine its exploratory, exploitative, local optima evasion, and convergence properties. Through quantitative and qualitative analyses, comparing MOHO with six prominent algorithms based on four significantly recognized performance measures, we demonstrated its effectiveness in handling real-world truss structure

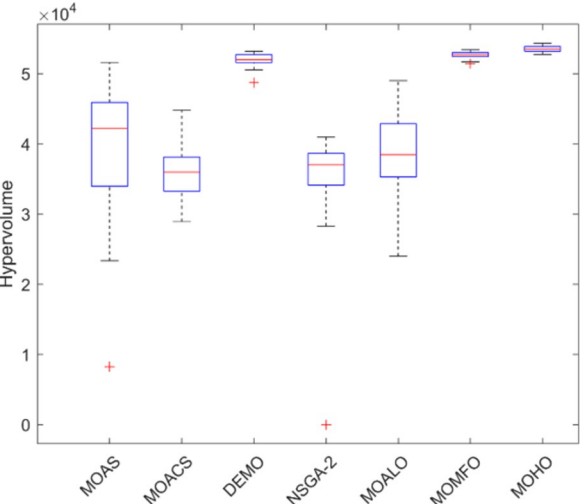

**Fig 31. Boxplots of 10-bar truss.**

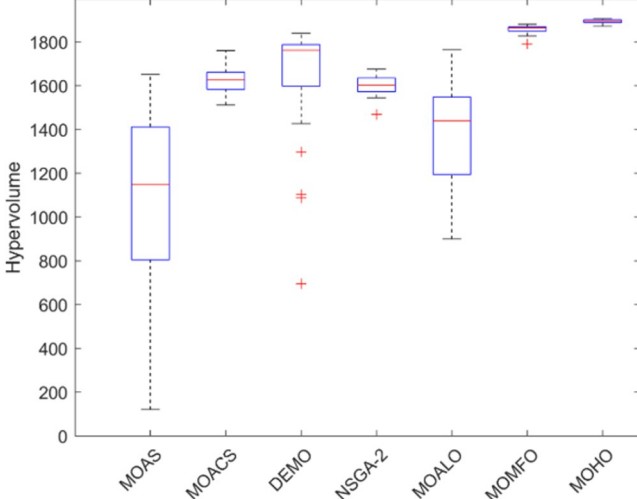

**Fig 32. Boxplots of 25-bar truss.**

optimization problems. Our results show that MOHO ranks first in the average Friedman rank test and outperforms alternative optimizers on all structural issues. MOHO demonstrated significant advantages concerning coverage, convergence, and solution diversification.

More research into how it performs on higher-dimensional engineering design challenges is essential to evaluate MOHO's potential thoroughly. Further studies could examine how MOHO can be applied to multi-modal and multi-dimensional technological problems with conflicting objectives. Additionally, the research can be expanded to investigate methods for enhancing performance and carrying out evaluations compared to other well-known optimization techniques. Further advances in efficiency and scalability will enable the MOHO for truss structures to manage more prominent and intricate structural systems. Promising directions include investigating its adaptability to different structural types beyond trusses, integration with advanced analysis techniques, and resilience in managing unknown parameters. The applicability and relevance of MOHO are further expanded in various engineering areas by

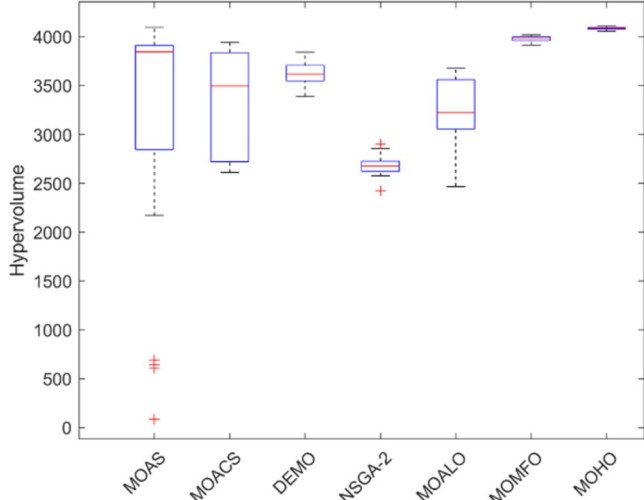

**Fig 33. Boxplots of 60-bar truss.**

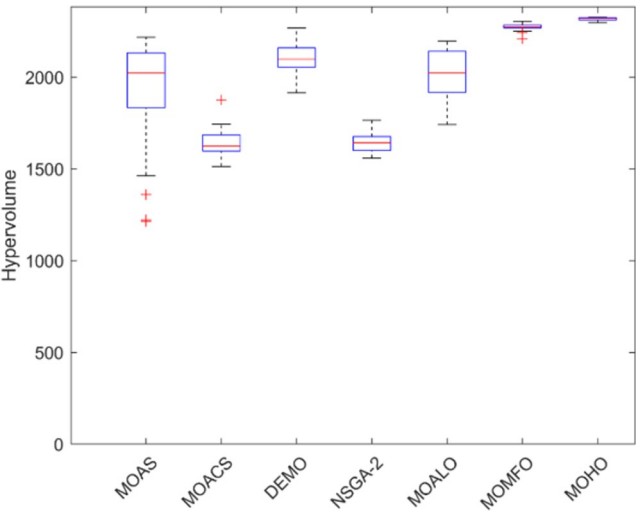

**Fig 34. Boxplots of 72-bar truss.**

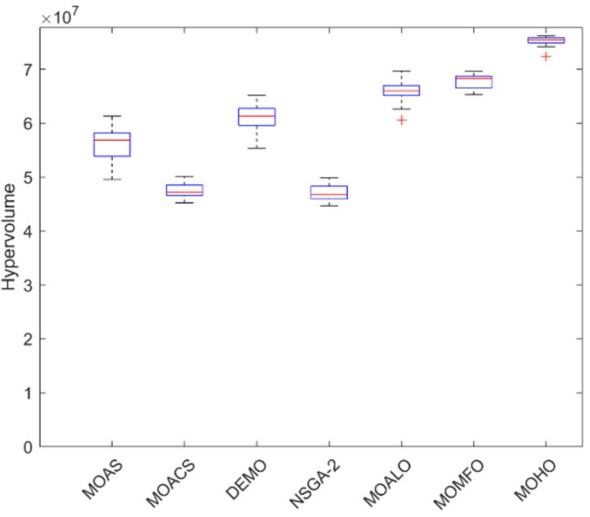

**Fig 35. Boxplots of 942-bar truss.**

**Table 13. The average and overall Friedman ranking test of all considered structures.**

|  | MOAS | MOACS | DEMO | NSGA-2 | MOALO | MOMFO | MOHO |
|---|---|---|---|---|---|---|---|
| 10-bar | 4.9500 | 4.8333 | 2.6333 | 6.2000 | 4.4083 | 2.5917 | **2.3833** |
| 25-bar | 5.6708 | 4.1083 | 3.3333 | 5.7250 | 4.4500 | 2.4875 | **2.2250** |
| 60-bar | 4.3500 | 4.1750 | 3.9000 | 5.1333 | 4.8667 | 2.8167 | **2.7583** |
| 72-bar | 4.3500 | 5.1750 | 3.5917 | 6.2833 | 4.0833 | 2.2167 | **2.3000** |
| 942-bar | 4.6167 | 5.4917 | 3.0833 | 6.7167 | 3.5250 | 2.8083 | **1.7583** |
| Average Friedman | 4.7875 | 4.7567 | 3.3083 | 6.0117 | 4.2667 | 2.5842 | **2.2850** |
| Overall Friedman rank | 6 | 5 | 3 | 7 | 4 | 2 | **1** |

hybridization with complementary optimization approaches and application to real-world engineering challenges. Enhancements in the future could involve extending MOHO to handle more extensive and more intricate truss structures containing increased design variables and constraints and additionally, adapting to dynamic environmental changes in design requirements using adaptive parameters. Overall, MOHO exhibits potential as a proficient and successful method for multi-objective optimization in truss-bar design issues, and its broader applicability to optimization contexts demands more research and analysis.

## Author Contributions

**Conceptualization:** Ghanshyam G. Tejani.

**Data curation:** Ghanshyam G. Tejani.

**Formal analysis:** Nikunj Mashru, Ghanshyam G. Tejani.

**Investigation:** Ghanshyam G. Tejani, Pinank Patel.

**Methodology:** Nikunj Mashru, Ghanshyam G. Tejani, Pinank Patel, Mohammad Khishe.

**Project administration:** Ghanshyam G. Tejani.

**Software:** Ghanshyam G. Tejani.

**Supervision:** Ghanshyam G. Tejani.

**Validation:** Nikunj Mashru, Pinank Patel, Mohammad Khishe.

**Visualization:** Nikunj Mashru, Pinank Patel, Mohammad Khishe.

**Writing – original draft:** Nikunj Mashru, Ghanshyam G. Tejani, Pinank Patel.

**Writing – review & editing:** Mohammad Khishe.

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
