## [Decision Letter · Decision Letter 0]

16 Jun 2024

PONE-D-24-17050Optimal Truss Design with MOHO: A Multi-Objective Optimization PerspectivePLOS ONE

Dear Dr. Khishe,

Thank you for submitting your manuscript to PLOS ONE. After careful consideration, we feel that it has merit but does not fully meet PLOS ONE’s publication criteria as it currently stands. Therefore, we invite you to submit a revised version of the manuscript that addresses the points raised during the review process.

We look forward to receiving your revised manuscript.

Kind regards,

Salar Farahmand-Tabar

Academic Editor

PLOS ONE

Reviewers' comments:

Reviewer's Responses to Questions

**Comments to the Author**

1. Is the manuscript technically sound, and do the data support the conclusions?

Reviewer #1: Yes

Reviewer #2: Yes

2. Has the statistical analysis been performed appropriately and rigorously? 

Reviewer #1: Yes

Reviewer #2: Yes

3. Have the authors made all data underlying the findings in their manuscript fully available?

Reviewer #1: Yes

Reviewer #2: Yes

4. Is the manuscript presented in an intelligible fashion and written in standard English?

Reviewer #1: No

Reviewer #2: Yes

5. Review Comments to the Author

Reviewer #1: Title of the manuscript: “Optimal Truss Design with MOHO: A Multi-Objective Optimization Perspective”

Summary: The authors propose a new metaheuristic inspired by the movement of hippos to solve multi-objective structural optimization problems from benchmark problems.

1 – The authors use the term “nodal deviation” in the abstract. Could this be more accurately described as “nodal displacement”?

2 - The authors also write size restrictions in the abstract. In optimization, the usual is to write “constraints” instead of “restrictions”.

3 – Please change “segment” to “section” in the paper's outline.

4 - Please check that all variable descriptions are correct in Tables 1, 2, and 3. Some variables are written superimposed on each other, for example, in Stages 1, 2, 6, 8, and 14.

5 – The authors use the word fitness for the first time in Section 3.2. However, the fitness function has not yet been adequately defined.

6 – The standard formulation of the optimization problem is incorrect. It is not about maximizing the maximum displacement “ ” but about minimizing the maximum displacement. Still, using the symbol epsilon to denote displacements is not usual, but rather “u” or “delta”, for example. Epsilon is commonly used to define “deformations”.

7 - The authors wrote again: “The primary objective is to maximize the nodal deflection and minimize the total mass…”

8 - Is the fitness function the penalty function? The text must clearly define this.

9 – Please change the term “infringement” to “violation”. It is the most common in the context of optimization.

10 – What values are adopted for the parameters in Equation (2)? The variables C and C_i are used in the same Equation. And these are not adequately defined.

11 – It is not usual to write “The circumstances for a load” but, for example, “Load cases” or “loading conditions,” which was used in the text.

12 – “Performance Comparison: Among all the average HV values of the algorithms considered, MOHO exhibits the highest values for truss structures, with respective averages of 53518.79, 1894.07, 4087.08, 2317.36, and 75288799”. In this sentence in the second bullet of Section 5, it is necessary to emphasize which experiments obtained these values, although these results are clear in the tables.

13 – It is suggested that the best values in boldface be highlighted in Tables 9 to 13 to make it easier for the reader to see.

14 – NSGA II is not properly written in the figure captions.

15 – Please change structural deformation to structural deflection. Deformation is different from displacement.

16 - In the labels of several figures, p10bar was used, etc. It could be 10-bar.

The manuscript's topic is interesting, and the authors propose a novel bioinspired metaheuristic. However, it demands a rigorous and in-depth review of technical definitions, standardization of definitions, expressions, variables, vocabulary, English proofreading, etc. It is suggested that optimization problems with a greater degree of complexity will be explored in future work.

Reviewer #2: The results have important value, it is my opinion that the article should be published after the minor revision.

Sufficient information about the previous study findings is presented for readers to follow the present study rationale and procedures.

The authors make a systematic contribution to the research literature in this area of investigation.

The title is adequate for the content discussed in the manuscript, explanatory, brief, and strong.

The article is novel and original which covers the scope of the journal.

The technology applied and the performance review of the proposed design is demonstrated in an efficient way

If required, the authors are informed to justify with real-time experimental analysis in the result and discussion section.

This article contains material which significant information on the current area of research.

The overall structure of the article is well-organized and in a well-balanced manner.

The article was written with the minimum length necessary for all relevant information.

However, future enhancement is to be described in the conclusion section.

6. PLOS authors have the option to publish the peer review history of their article (what does this mean?). If published, this will include your full peer review and any attached files.

Reviewer #1: No

Reviewer #2: **Yes: **Dr B Santhosh Kumar

---

## [Author Response · Author response to Decision Letter 0]

25 Jun 2024

The authors have responded to all of the reviewers’ comments and have also addressed the points mentioned above. The manuscript has also been revised accordingly. We sincerely appreciate your insightful and constructive comments and suggestions. We believe that these will greatly strengthen the paper. We hope that you find revised focus helpful. Authors would be glad to make further revisions, if necessary for improving the overall quality of the manuscript. We much appreciate the Editor’s comments and the manuscript now incorporates all the suggestions as discussed earlier.

---

## [Decision Letter · Decision Letter 1]

25 Jul 2024

Optimal Truss Design with MOHO: A Multi-Objective Optimization Perspective

PONE-D-24-17050R1

Dear Dr. Khishe,

We’re pleased to inform you that your manuscript has been judged scientifically suitable for publication and will be formally accepted for publication once it meets all outstanding technical requirements.

Kind regards,

Salar Farahmand-Tabar

Academic Editor

PLOS ONE

Additional Editor Comments (optional):

Reviewers' comments:

Reviewer's Responses to Questions

**Comments to the Author**

1. If the authors have adequately addressed your comments raised in a previous round of review and you feel that this manuscript is now acceptable for publication, you may indicate that here to bypass the “Comments to the Author” section, enter your conflict of interest statement in the “Confidential to Editor” section, and submit your "Accept" recommendation.

Reviewer #1: All comments have been addressed

Reviewer #2: All comments have been addressed

2. Is the manuscript technically sound, and do the data support the conclusions?

Reviewer #1: Yes

Reviewer #2: Yes

3. Has the statistical analysis been performed appropriately and rigorously? 

Reviewer #1: Yes

Reviewer #2: Yes

4. Have the authors made all data underlying the findings in their manuscript fully available?

Reviewer #1: Yes

Reviewer #2: Yes

5. Is the manuscript presented in an intelligible fashion and written in standard English?

Reviewer #1: Yes

Reviewer #2: Yes

6. Review Comments to the Author

Reviewer #1: The authors have addressed all the reviewer's comments, and the manuscript should be accepted for publication.

Reviewer #2: The whole article is properly written understandably. Moreover, this article sounds well with various aspects of this research area and the involvement of this work is appreciable

7. PLOS authors have the option to publish the peer review history of their article (what does this mean?). If published, this will include your full peer review and any attached files.

Reviewer #1: No

Reviewer #2: **Yes: **Dr Santhosh Kumar Balan

---

## [Editor Report · Acceptance letter]

8 Aug 2024

PONE-D-24-17050R1 

PLOS ONE

Dear Dr. Khishe, 

I'm pleased to inform you that your manuscript has been deemed suitable for publication in PLOS ONE. Congratulations! Your manuscript is now being handed over to our production team.

Kind regards, 

on behalf of

Dr. Salar Farahmand-Tabar 

Academic Editor

PLOS ONE